# Direct contribution of the sensory cortex to the judgment of stimulus duration

Sebastian Reinartz [1,2,4], Arash Fassihi[1,3,4], Maria Ravera [1], Luciano Paz[1], Francesca Pulecchi[1], Marco Gigante[1] & Mathew E. Diamond [1] ✉

Decision making frequently depends on monitoring the duration of sensory events. To determine whether, and how, the perception of elapsed time derives from the neuronal representation of the stimulus itself, we recorded and optogenetically modulated vibrissal somatosensory cortical activity as male rats judged vibration duration. Perceived duration was dilated by optogenetic excitation. A second set of rats judged vibration intensity; here, optogenetic excitation amplified the intensity percept, demonstrating sensory cortex to be the common gateway both to time and to stimulus feature processing. A model beginning with the membrane currents evoked by vibrissal and optogenetic drive and culminating in the representation of perceived time successfully replicated rats' choices. Time perception is thus as deeply intermeshed within the sensory processing pathway as is the sense of touch itself, suggesting that the experience of time may be further investigated with the toolbox of sensory coding.

The neuronal mechanisms underlying the feeling of the elapsed time of sensory events remain unknown. If the neuronal substrate for perceived time is envisioned as a distributed network[1,2] rather than as a single restricted population dedicated to the function of time measurement, then a working definition for this network is the set of brain regions within which a variation in firing leads directly and systematically to a variation in perceived time. One series of studies addresses the role of the striatum[3–5]. There is also evidence for a role of cortical processing networks in time perception[6–8]. However, it is commonly believed that within cortical networks the *primary* sensory cortex merely relays start and stop signals to a central processing network. In disagreement with most current frameworks[9–12], our hypothesis is that the ongoing activity of the primary somatosensory cortex plays a direct and systematic role in the judgment of time. The neural substrate of time perception, we posit, might involve integration and accumulation of the drive within the sensory processing pathway[13], as suggested in human psychophysical experiments across various sensory modalities[14–18]. As such, it is useful to pinpoint which sensory processing structure participates in the percept. If the cortical sensory representation is one component of the substrate for the time percept,

then the detailed firing patterns within the primary sensory cortex will directly mediate the feeling of the passage of time. To support or refute this hypothesis, and to compare it to competing hypotheses, a quantitative, causal relationship between neuronal firing and time perception is needed.

Alongside the investigation of the role of sensory cortical firing in time perception, we examine the psychophysical effects of optogenetic intervention in a set of rats who judged the intensity of tactile stimuli. This control group provides a demonstration that, while manipulation of the tactile sensory cortex affords unexpected results in time perception, it also leads to expected effects on tactile perception. Somatosensory cortex, we will argue, functions within its sensory modality, and beyond it.

In sum, here we seek to determine whether the tactile sensory cortex is part of the neuronal substrate for perceived time and, if so, what are the features of firing that causally lead to shifts in perceived time. The coding algorithms for sensory features are well established[19–25]. Guided by these algorithms, we tested whether the effects of optogenetic manipulation on duration judgment could be predicted using real neuronal spiking patterns as input. The successful

¹SENSEx Lab, International School for Advanced Studies (SISSA), 34136 Trieste, Italy. ²Present address: Brain & Sound Lab, Department of Biomedicine, Basel University, 4056 Basel, Switzerland. ³Present address: Department of Physics, University of California, San Diego, La Jolla, CA 92093, USA. ⁴These authors contributed equally: Sebastian Reinartz, Arash Fassihi. ✉e-mail: diamond@sissa.it

implementation of a computational framework for the perceived duration of tactile stimuli based on the firing patterns evoked by those stimuli opens up the field of time perception to the tools of sensory coding.

## Results

The subjective experience of an external stimulus has a dual nature – the feeling of the physical features of the sensory input and, in parallel, the feeling of the time occupied by that stimulus[13]. While decades of research have built an understanding of the basic neuronal coding algorithms for stimulus features[19–25], a mechanistic, causal understanding of the percept of the elapsed duration of an event is still lacking. Here, we combine rat psychophysics with optogenetics to demonstrate that perception of stimulus duration may be treated with the language of sensory coding.

### Duration and intensity percepts interact

On each trial, rats compared two vibrissal vibrations (stimulus 1, stimulus 2; Fig. 1a). Vibrations were constructed by concatenating a sequence of speed values, sampled from a half Gaussian distribution[26]. A single vibration was defined by its intensity ($I$) in units of mean speed, and its duration ($T$). We trained two sets of rats. Duration rats had to compare the two stimuli according to their relative time spans ($T1 > T2$ or $T2 > T1$) and select the associated reward spout. Intensity rats had to compare the two stimuli according to the analogous relation ($I1 > I2$ or $I2 > I1$). The two groups received the same stimulus set (Supplementary Fig. 1), the only difference being the feature they were trained to extract – for duration rats, stimulus intensities were irrelevant to reward location, while for intensity rats stimulus durations were irrelevant (Fig. 1a, gray and red arrows prior to choice).

In the upper plot of Fig. 1b, the left bar depicts the performance (78% correct) of duration rats when choices are analyzed according to relative stimulus durations. When the same choices are analyzed according to relative stimulus intensities (right bar), performance was above chance (54% correct). Intensity rats performed at 85% correct according to relative stimulus intensities (lower plot, right bar); when analyzed according to relative stimulus durations (left bar), performance was above chance (53% correct). The 3–4% deviation from 50% when choices are analyzed according to the untrained feature indicates that this feature influences judgments that would, ideally, be based on only the relevant feature[13].

We further examined the effects of the relevant and irrelevant features. The psychometric curves of Fig. 1c show choices according to graded stimulus differences. In the upper plot, choices in duration rats (gray) were governed by $\Delta T$ (normalized duration difference, defined as $(T2 - T1)/(T2 + T1)$) while choices in intensity rats (red) were weakly modulated by $\Delta T$. In the lower plot, choices in intensity rats (red) were governed by $\Delta I$ (normalized intensity difference, $(I2 - I1)/(I2 + I1)$) while choices in duration rats (gray) were weakly modulated by $\Delta I$.

To quantify the bias caused by the irrelevant feature (intensity) on duration perception, we computed how the average incidence of judging $T2 > T1$ was affected by $\Delta I$ (Fig. 1d, upper plot; also see Methods). Similarly, we quantified the duration-dependent bias in perceived intensity by computing how the average incidence of judging $I2 > I1$ was affected by $\Delta T$ (Fig. 1d, lower plot).

While earlier work[5], using a data set partially overlapping that of the present study, demonstrated the effect of the irrelevant feature through overall performance (as in Fig. 1b) and by choice probabilities, the psychometric curves (Fig. 1c) and the bias measure (Fig. 1d) given in the current analysis are novel.

### Optogenetic control of perception

The systematic interaction between perceived duration and perceived intensity (Fig. 1a–d) leads to the hypothesis that the neuronal representation of stimulus features – here, vibration intensity – might

constitute the basis of some forms of time perception. The remaining experiments test this hypothesis by trying to specify an underlying neuronal code within vibrissal somatosensory cortex (vS1) that could account for rats' judgment of both features. ChR2(H134R) was expressed in left vS1 (Fig. 2a, left) of 5 rats and neuronal populations were accessed by movable microdrive arrays coupled with optic fibers (Fig. 2a, middle). If vS1 directly participates in generating time judgments, optogenetic excitation of vS1 (Fig. 2a, right panel, lower left data) will systematically bias the psychometric curves. Specifically, boosting the firing evoked by the tactile stimulus will cause that stimulus to be perceived as occupying a longer period. As a control, in 2 duration-trained rats eNpHR3.0, was expressed in left vS1 (Fig. 2a, right panel, lower right data). Hereafter, optogenetic interventions in ChR2(H134R)-expressing rats and in eNpHR3-expressing rats are referred to as photoexcitation and photoinhibition, respectively.

The principal findings are shown in Fig. 2b. The effects of photoexcitation and photoinhibition during stimulus 2 of the delayed comparison task are compared to the no-light condition (left panel). The psychometric curves associated with these conditions, pooled across rats and sessions, are given in Fig. 2b, middle panel. Under photoexcitation, the psychometric curve shifted leftward, indicating an overestimation of stimulus 2 duration, as compared to the no-light condition (Fig. 2b, middle, light blue versus black). The control condition, photoinhibition, gave a weak but significant rightward shift (red versus black). If the effects of blue light application in ChR2(H134R)-expressing rats were due to tissue heating, one would also expect to find tissue-heating effects with red light application in eNpHR3.0-expressing rats; instead, red light appeared to yield an effect consistent with opsin-mediated neuronal inhibition.

The right panel shows the opposing perceptual biases of photoexcitation versus photoinhibition during stimulus 2, visualized by plotting the psychometric curve point of subjective equality (PSE) against the percent of trials where $T2$ was judged longer than $T1$, across different values of $\Delta T$ (see Methods). Notwithstanding individual differences in the magnitude of effect, the separation between the blue and red distributions signifies that rats were biased towards judging stimulus 2 as having longer duration in the photoexcitation condition, as compared to the photoinhibition condition.

Rats showed the same acuity in detecting duration differences on light-on versus no-light trials, as assessed by the slopes (quantified as the inverse of the psychometric parameter $\sigma$) of the three psychometric curves of the middle panel (vS1 photoexcitation versus no-light trials, $p = 0.93$; photoinhibition versus no-light trials, $p = 0.23$; resampling method, permutation test, 1000×). Thus, while optogenetic intervention caused changes in perceived time, the altered percept was reliable (see also Supplementary Fig. 2b, c).

Photoexcitation during presentation of stimulus 2 caused a leftward shift of the psychometric curve, indicating an overestimation of that vibration's duration, as compared to the shift obtained with photoexcitation during stimulus 1 (Fig. 2c; see also Supplementary Fig. 3 for the individual rat psychometric curves). Estimation of the fitting parameters (Supplementary Fig. 4) indicates that changes in psychometric curves are explained by a horizontal shift ($\mu$) and not by changes in acuity ($\sigma$) or lapse rates ($\gamma, \lambda$).

Three additional experiments further support the argument that vS1 firing is a key ingredient in perceiving stimulus duration. EYFP-ChR2(H134R)-expressing rats ($n = 4$) trained to compare the duration of each trial's single vibration duration to a fixed, reference duration showed a bias towards longer perceived duration on trials with photoexcitation (Supplementary Fig. 5). Here the light was modulated stochastically across the tactile stimulus presentation. This finding indicates that the vS1 role in stimulus duration perception is not specific to the delayed comparison working memory task, nor is it specific to the profile of light application in optogenetic trials. Additionally, blue light presented above the apparatus, applied with

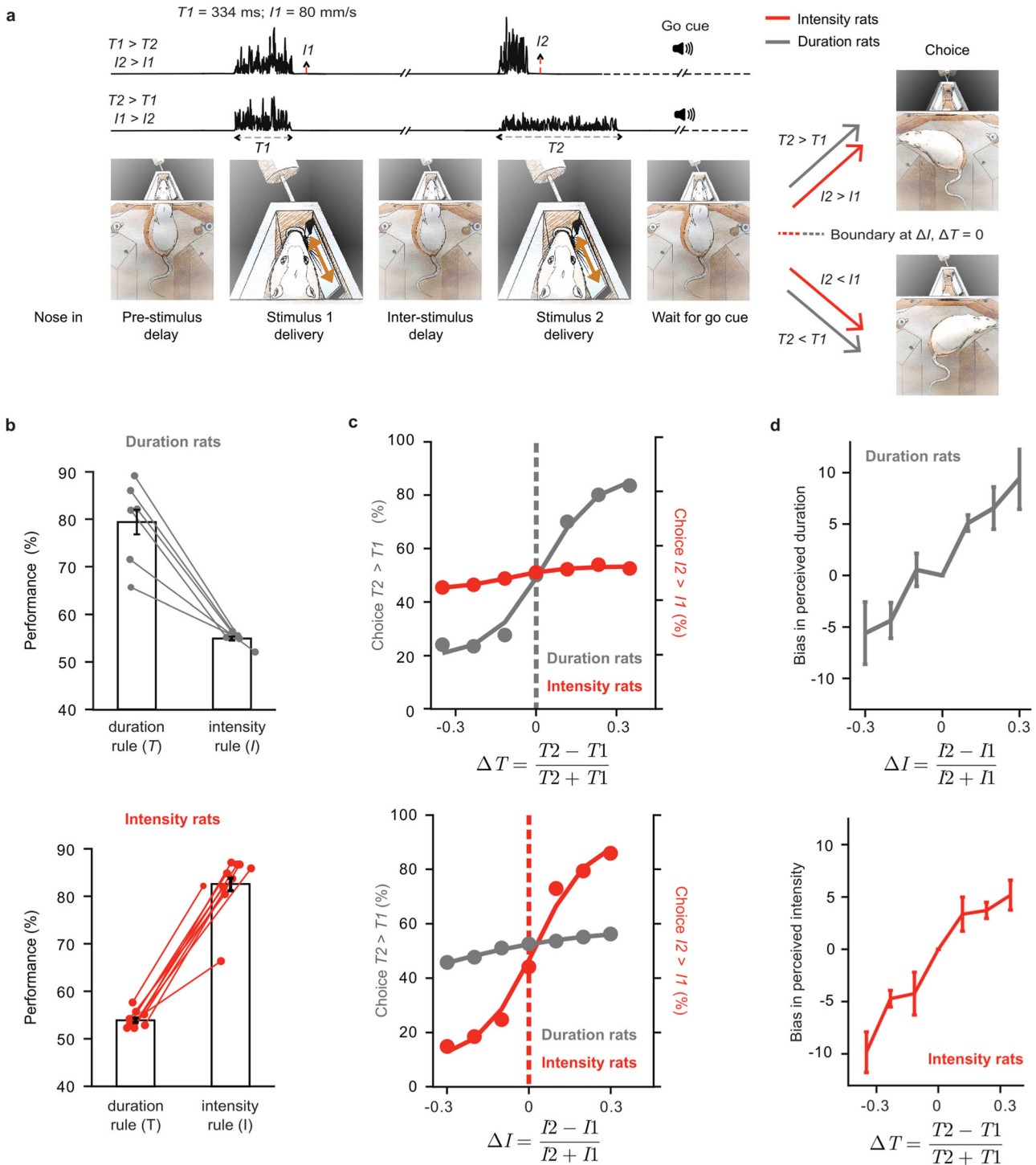

**Fig. 1 | Interacting perception of duration and intensity. a** Rat enters nose poke, bringing its right whiskers into plate contact. Following a pre-stimulus delay (0.5 s), stimulus 1 is delivered through plate motion. Stimulus 2 is presented after the 2-s inter-stimulus delay. Acoustic go cue prompts the rat to make a choice. **b** Upper: performance of duration rats (*n* = 6) with choices computed according to duration and intensity rules. Lower: performance of intensity rats according to duration and intensity rules (*n* = 10). Analysis of choices according to duration rule done on trials with largest Δ*T*; analysis of choices according to intensity rule done on trials with largest Δ*I*. Each dot represents a single subject. **c** Upper: Psychometric curves of both sets of rats based on Δ*T*. Lower: Psychometric curves of both sets of rats based on Δ*I*. Both plots are averaged across all subjects. **d** Upper: Intensity-dependent bias in perceived duration. For a given Δ*I*, bias is the average of the percent of trials judged *T2 > T1* across Δ*T* values. Lower: Duration-dependent bias in perceived intensity, computed in the analogous way. Bias measure details in Methods. Standard error of the mean (SEM) across animals (upper: *n* = 6 rats, examined over 531 independent experiments; lower: *n* = 10 rats, examined over 878 independent experiments) indicated as error bars. Part of the behavioral data presented here reanalyzed from an earlier study[5] (all 6 duration rats shown in their Fig. 1c, and 7 of 10 intensity rats shown in their Figure 6B). Source data are provided as a Source Data file.

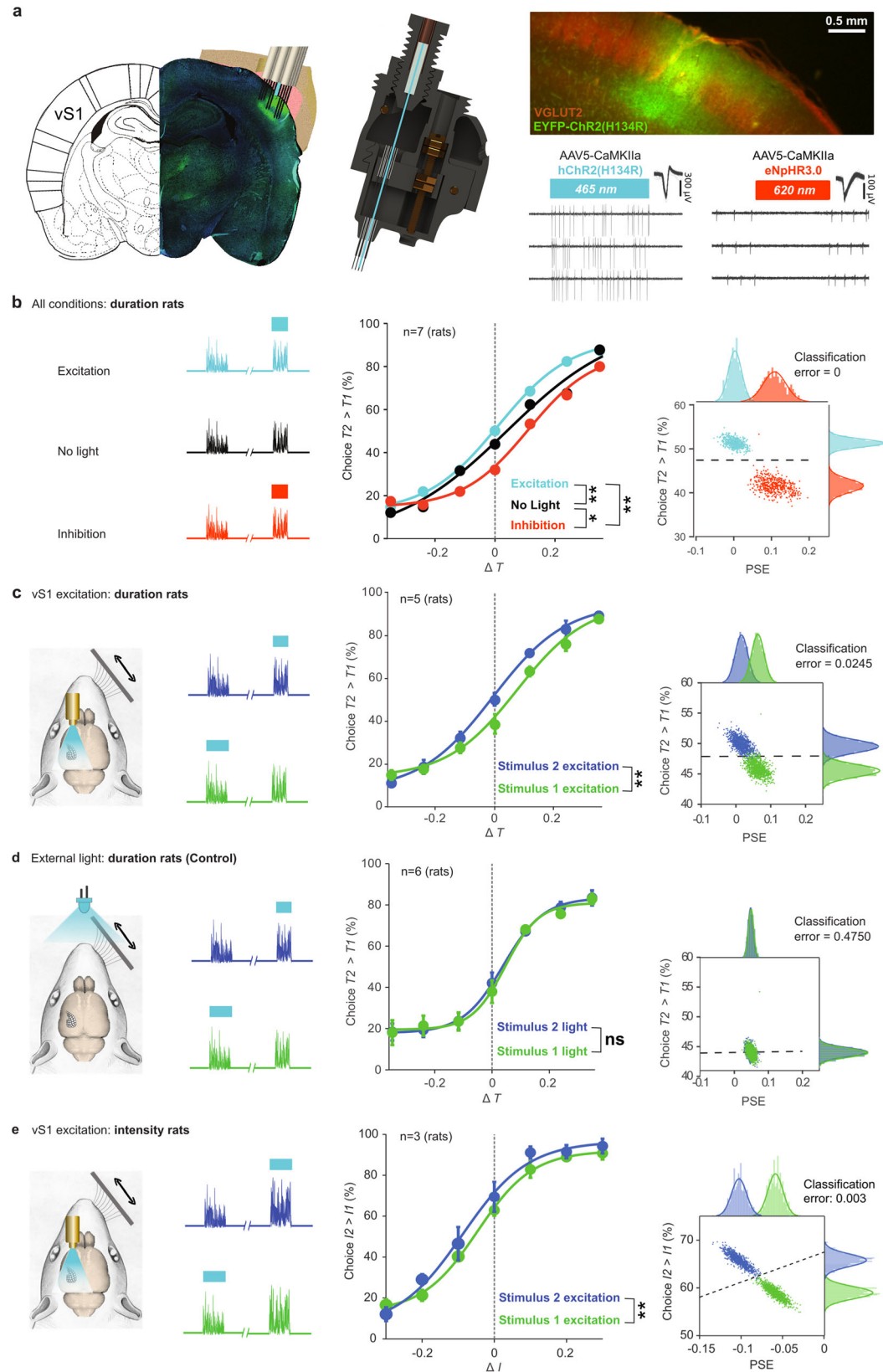

the same temporal alignment to the vibration, had no effect (Fig. 2d), indicating that behavioral biases were not likely to be due to visual cues. The intensity bias of Fig. 1 (*T2 > T1* choice more likely when *I2 > I1*) was conserved across all three experimental conditions (photoexcitation, control, and photoinhibition), indicating that optogenetic intervention acted within the regime of natural coding

and decoding (Supplementary Fig. 6). One final control experiment indicates that the vS1 neuronal population involved in building the duration percept also participates in the perception of tactile stimulus features. In intensity rats expressing EYFP-ChR2(H134R), optogenetic excitation during presentation of stimulus 2 or stimulus 1 caused overestimation of that vibration's intensity (Fig. 2e).

**Fig. 2 | Optogenetic manipulation of time and intensity perception. a** Left: EYFP-ChR2(H134R)-injected brain with optical fiber surrounded by electrode array. Middle: custom-built multisite drivable optrode array. Upper right: coronal section of EYFP-ChR2(H134R) injection site (green) counterstained with Anti-Vglut2 primary antibody (red). Lower right: traces of two vS1 single-neurons. In the EYFP-ChR2-injected rat 465 nm illumination (blue bar, 500 ms) excited the neuron while in the eNpHR3.0-injected rat 620 nm illumination (red bar, 250 ms) inhibited the neuron. **b** Left and middle: compared to no-light condition (black), photoexcitation (blue) and photoinhibition (red) during stimulus 2 yielded, respectively, an increased and decreased likelihood of judging $T2 > T1$. Statistical significance in the middle panels of (**b–e**) is evaluated by resampling (one-sided permutation test, 1000x), subtracting at each iteration the averaged percentage of choice (excluding easy trials) between the respective condition pair (see Methods). p-values labels are *$p < 0.05$; **$p < 0.01$; ns for $p > 0.05$. Exact values are: Chr2 vs eNpHR3.0, $p < 0.001$, Chr2 vs control, $p = 0.003$, eNpHR3.0 vs control, $p = 0.009$. Right: two signatures of curve shift, percent of trials judged $T2 > T1$ irrespective of stimulus duration (ordinate) and point of subjective equality (PSE) (abscissa), measured with bootstrap resampling (1000×). A support vector machine classifier quantifies data separation by classification error. **c** Left and middle: excitation during stimulus 2 (blue) compared to stimulus 1 (green). Green vs blue difference significant, $p = 0.005$. Left vibrissae were intact but are omitted in all sketches. SEM across individual rats indicated as error bars. Statistics derived across $n = 5$ rats, examined over 113 independent experiments. **d** Left: control condition with external LED (465 nm) illuminated above the rat with same wavelength and timing as optogenetic light delivery in photoexcitation trials. Middle: light-on during stimulus 2, compared to stimulus 1 (green) revealed no duration perceptual bias in relation to visual cues ($p = 0.09$). SEM across individual rats indicated as error bars. Statistics derived across $n = 6$ rats, examined over 202 independent experiments. **e**. Same as **c**, but for intensity rats ($p < 0.001$). Statistics derived across $n = 3$ rats, examined over 37 independent experiments. Source data are provided as a Source Data file.

In each of the experiments illustrated in Fig. 2, optogenetic intervention had no choice effect when the two stimuli differed substantially in the sensory feature to be compared, namely, when $\Delta T = -0.35$ or $0.35$ for duration rats and when $\Delta I = -0.3$ or $0.3$ for intensity rats. If optogenetic manipulation had disrupted performance directly at the decisional level, it would be expected to affect choices no matter which stimulus pair was presented; instead, the manipulations merely shifted the percept in a systematic way, as borne out by the psychometric parameters (Supplementary Fig. 4). Taken together, the results point to a multiplexing of information within vS1, where neuronal activity appears to be used to construct percepts both within the tactile modality (vibration intensity) and beyond the tactile modality (elapsed time).

## Time coding

Through what physiological mechanisms does the neuronal firing within vS1 contribute to the percept of the passage of time? Fig. 3a illustrates two example neuronal clusters from duration-trained EYFP-ChR2(H134R)-expressing rats, representative of the population's heterogeneity in responsiveness to vibrissal stimulation and photoexcitation[27]. The neuronal cluster in the left plots responded weakly to vibrissal stimulation, showing only an onset transient. The same cluster was robustly excited by blue light. The neuronal cluster in the right plots gave a strong, non-adapting response to vibrissal stimulation, but was not excited by blue light. Population sensory responses, with and without optogenetic excitation, are shown in Fig. 3b. Color indicates the deviation of firing rate from the baseline level (z-score; see Methods for definition of baseline). In the upper plot, neuronal clusters are ordered by response magnitude to stimulus 2, which was accompanied by photoexcitation. In the lower plot, neurons are ordered by response magnitude to stimulus 1, which was accompanied by photoexcitation. The population's response to vibrissal stimulation alone and vibrissal-plus-photoexcitation can be seen by comparing the vertically aligned upper and lower plots (quantification in Supplementary Figs. 7, 8).

The temporal profile of vS1 sensory responses was conserved under photoexcitation (similarly to the conserved stimulus-evoked profiles in auditory cortex[28]). The conserved sensory response profile offers a physiological measure to help explain why optogenetic intervention did not disrupt the fundamental psychophysical parameters of time judgment, apart from perceived duration (i.e., Fig. 2, Supplementary Figs. 2b, c, 4). Figure 3c, left, shows the population peristimulus time histogram (PSTH) in EYFP-ChR2(H134R)-expressing rats in the absence of photoexcitation. To allow pooling across different durations, the first 150 ms and final 50 ms are shown. After an early peak, firing rate remained stable until offset. The right panel shows the PSTHs of the same population under photoexcitation, revealing a modest boost in the vibrissae-evoked response, particularly at vibration onset and thereafter distributed evenly across the period of tactile stimulation.

To evaluate the role of vS1 in the two behavioral tasks, we first consider the control rats trained to compare vibration intensities. Figure 3d illustrates the vS1 population ($n = 138$ neurons) mean firing rates recorded in EYFP-ChR2(H134R)-expressing rats with light-off (abscissa) and light-on (ordinate); colors denote vibration intensity. In agreement with earlier work suggesting a vS1 role in the neuronal representation of vibration intensity[13,20], firing rates were higher for greater intensity vibrations. Further analysis is given in Supplementary Figs. 9, 10. Projection along the diagonal reveals the overall effect of photoexcitation – a leftward shift towards a higher firing rate (depicted for the 64 mm/s intensity, green). Thus, if firing rate is positively correlated with vibration intensity, the higher firing rate with photoexcitation explains the bias towards stronger perceived intensity (Fig. 2e).

Next, we consider the rats trained to compare vibration durations. If firing rate functions as an explicit representation of stimulus duration within vS1 (as it does for intensity), it must vary systematically in relation to the passage of time. Figure 3e examines the firing rate of the entire recorded vS1 population during the final 100 ms of stimulus presentation. Points were obtained by bootstrap resampling and colors denote stimulus duration. Projection along the diagonal reveals the overall effect of optogenetic excitation – a leftward shift towards higher firing rate (depicted for the 334 ms duration, green). However, when the points are projected laterally and vertically, the marginal distributions for each duration are fully overlapping. Thus, firing rate at the end of the stimulus (like the firing rates across the entire stimulus duration, see Supplementary Fig. 11), although boosted by the optogenetic intervention, did not vary in relation to duration and would not provide a robust code.

As an alternative, Fig. 3f examines the spike count as the possible basis for a duration code. Following the format of the preceding panel, we computed counts from vibrissal-stimulus onset to offset, with light-off (abscissa) and light-on (ordinate). The projection along the diagonal reveals a leftward shift towards greater spike count under photoexcitation (again, depicted in green for the 334 ms duration; note the scale bar of 5 spikes). Differently from the rate code, when points are projected laterally and vertically, marginal distributions separate according to stimulus duration. Thus, summated spike count could provide a downstream integrator with an input that robustly maps to duration and is consistent with the dilation of perceived time generated by optogenetic excitation.

To ascertain the neuronal signal that is available to downstream regions, it is informative to first consider the magnitude of the optogenetic effect in linear terms. By plotting how much time from stimulus onset must have passed in trials with light-off and light-on to reach any selected count of spikes, Fig. 3g shows the signal available in the linear sum of vS1 spikes. The fact that points fall below the solid diagonal indicates that a given spike count was reached earlier when optogenetic excitation was applied. For instance, when 20 spikes (data

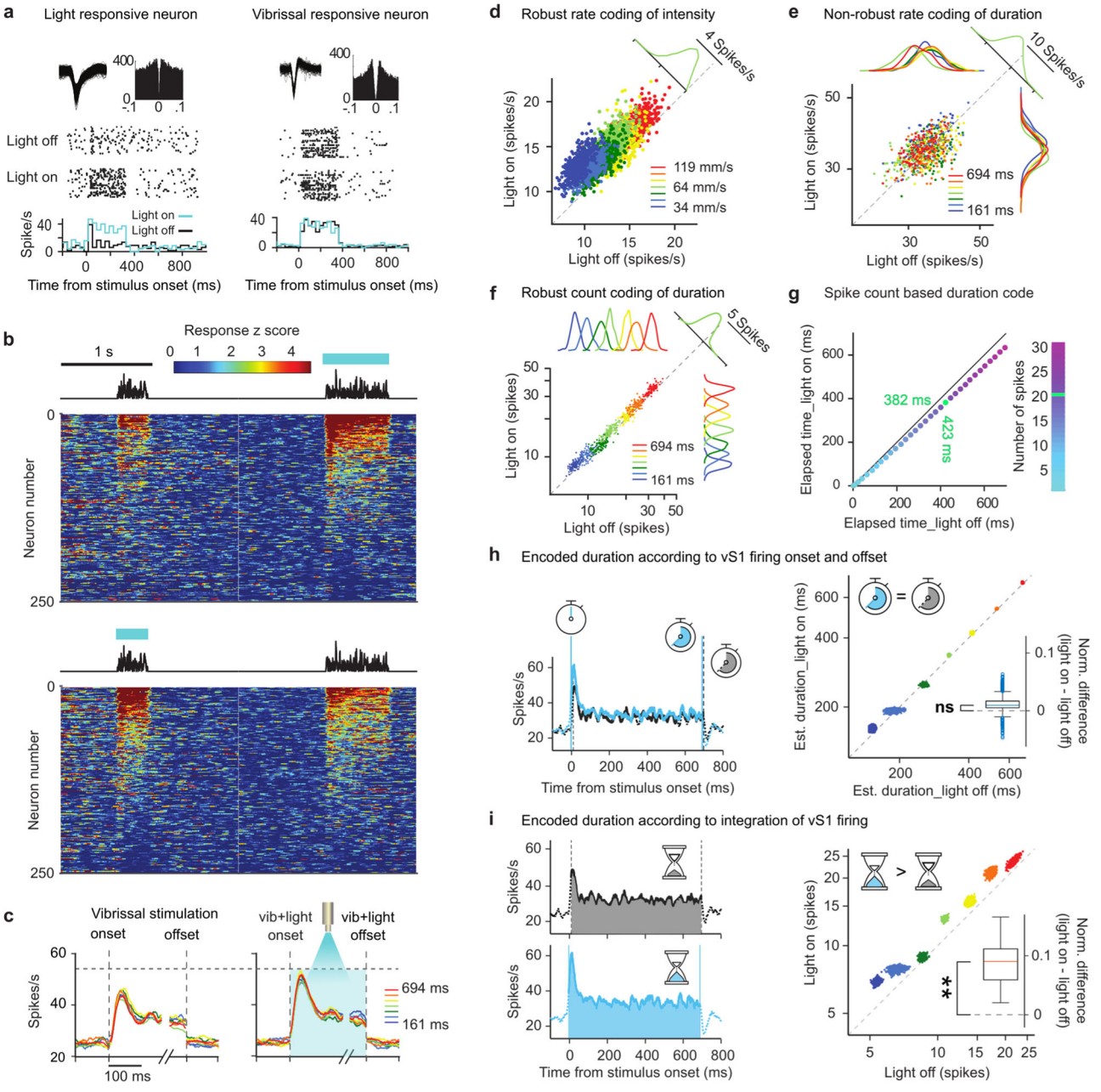

**Fig. 3 | vS1 coding of duration. a** Example vS1 recordings. Upper: spike waveforms and spike time autocorrelogram. Middle: raster plots of randomly selected 334-ms light-off and light-on trials. Lower: corresponding firing rates in non-overlapping 20-ms bins, light-on (blue) and light-off (black) trials. Single-and multi-units (right, left examples, respectively) pooled for analysis. **b** Normalized response of neurons (*n* = 250) on trials with stimulus 1 and 2 of 334- and 694-ms duration, respectively; intensity 64 mm/s. Above the response plots, example vibrations (plate speed across time) are illustrated as black lines. **c** Average response (as PSTH; see Methods) with light-off (left) and light-on (right). Light onset/offset matched vibrissal onset/offset. **d** Each dot shows population mean firing rate colored by vibration intensity in EYFP-ChR2(H134R)-expressing intensity rats (*n* = 3). In the bootstrap resampling algorithm[13], the population (*n* = 138 neurons) is resampled to include 90% coding and 10% non-coding neurons (see Methods) such that the photoexcitation-evoked firing rate increase resembles the psychometric effect. Dashed diagonals in (**d, f**) denote equal firing for light-on and light-off. **e** Each dot shows population mean firing rate (bootstrap resampling) colored by vibration duration. Distributions shown as marginals. For 334-ms duration, points are projected parallel to the diagonal to give the green histogram. **f** Same as (**e**), but for spike count summated across the entire stimulus presentation. **g** Points (obtained

by resampling) depict elapsed time to reach a given spike count (see color scale), light-off vs light-on. **h** Left: light-off (black) and light-on (light blue) population PSTH in response to 694-ms vibrations; 1 ms bins with 15 ms smoothing. Detected response onset and offset given by left and right vertical lines. Right: Points (obtained by resampling) colored by actual vibration duration (**f**), show stimulus duration estimated by elapsed onset-to-offset time of evoked activity with light-off and light-on. Diagonal denotes equal duration for light-on and light-off. Inset: Normalized difference in estimated duration (light-on − light-off) across all durations fails to uncover any significant photoexcitation-evoked shift (*p* = 0.147). **i.** Left: same as (**h**), for integration hypothesis. Right: Points (obtained by resampling), colored by vibration duration, show spike count summated from onset to offset. Inset: Normalized difference in onset-to-offset spike counts (light-on − light-off) across all durations uncovers a significant photoexcitation-evoked shift (*p* = 0.006). **h, i** use the color code of (**c**). Statistics for (**h, i**) derive from 1000x resamples from 138 neurons. Box plots (Matlab function *boxchart*): the central line is the sample median, the edges of the box are the 25th and 75th percentiles, whiskers extend to the most extreme data points not counting outliers, and the outliers are plotted as circles. One-sided permutation test (1000×) applied for statistical significance. Source data are provided as a Source Data file.

point in green) have been integrated, 423 ms would have passed on trials with light-off, but just 382 ms on trials with light-on. A corresponding quantification of the optogenetic effect of vS1 spikes in intensity perception is shown in Supplementary Fig. 10.

One plausible mechanism by which vS1 might contribute to the perception of elapsed stimulus time is by providing downstream centers with onset and offset times that could function similarly to a stopwatch. The increase in perceived duration with photoexcitation, in this scenario, would result from a greater elapsed time interval between onset and offset. This model is tested in Fig. 3h. Onset and offset of the vibration-evoked responses were detected when the firing rate crossed a threshold, set as 2.2 STD of change from the baseline firing rate (details in Methods). In the left panel, neuronal population onset and offset times for vibrations of 694 ms are shown as vertical lines under the light-on condition (blue) and light-off (black) conditions. Nearly identical temporal boundaries are recovered under the two conditions, symbolized by the blue and gray stopwatches which record the same elapsed time. The right panel gives the overall result of the test for whether start-to-end elapsed time can explain perceived time dilation in the condition of photoexcitation. For all examined vibration durations, elapsed time under the light-off (abscissa) and light-on (ordinate) conditions lay along the diagonal. The inset shows the normalized difference between estimated elapsed time under the light-on and light-off conditions. Thus, the time between the onset and offset of vibration-evoked neuronal firing was not significantly affected by optogenetic intervention (again symbolized by the blue and gray stopwatches), indicating that this model does not offer a straightforward account for duration over-estimation in the photoexcitation condition.

The alternative model, where perceived duration is related to the integration of vS1 firing, is given in Fig. 3i. The plots of the left panel illustrate the firing feature that could underlie this coding mechanism, the total (filled) area under the PSTH. By boosting vS1 excitability, the total number of spikes transmitted to downstream integrators, for any given actual vibration duration, is increased under photoexcitation. This accumulation of vS1 activity is symbolized by the hourglasses, where the readout records a greater duration with light-on (lower plot) versus light-off (upper plot). In the right panel, the accumulated spikes between response onset and offset are plotted under the light-off (abscissa) and light-on (ordinate) conditions. Similarly to the plot in Fig. 3f, greater accumulated spike counts between estimated onset and offset are seen for any given stimulus duration under the light-on condition. As in the left panel, this accumulation of vS1 activity is symbolized by the two hourglasses.

## Model of coding mechanisms

The preceding analyses represent preliminary tests of the hypothesis that the tactile sensory cortex contributes to the neuronal substrate for perceived time not by an explicit firing rate code (Fig. 3e) nor by relaying stimulus onset/offset (Fig. 3h), but through an integrative mechanism (Fig. 3f, g, i). These initial tests treated the sensory cortex as making a linear contribution to the final duration percept, as vS1 spiking output is progressively accumulated. However, earlier studies indicate that the dynamics of downstream integration underlying the perception of stimulus duration (and stimulus intensity, not the object of the present study) are likely to be nonlinear[13,20]. Therefore, we grounded the search for mechanisms in a 3-stage model encompassing non-linear integration. In stage 1, vibrissal drive and optogenetic drive evoke currents in vS1 neurons, leading to spiking through a linear-nonlinear Poisson (LNP) process (Fig. 4a, left). After finding the parameters that produce simulated spike trains mimicking the original spike trains (stimulus-dependence, variability, and diversity), we recombine these currents in a Gaussian Mixture Model[29] to create a large pool of simulated vS1

neurons (Supplementary Figs. 12, 13). In stage 2, the accumulated quantity ($\Upsilon$) in a leaky integrator (LI) downstream to vS1 is taken as the duration percept. As the integrator summates incoming spikes, input continuously leaks out by some proportion ($\Upsilon/\tau$) (Fig. 4a, upper right). In stage 3, the values of $\Upsilon$ at the conclusion of stimuli 1 and 2 are taken as the explicit readouts of duration and their comparison predicts the rat's actual choice (Fig. 4a, lower right).

This model yields the neurometric curves (solid lines) of Fig. 4b, which overlie the observed psychometric data (points), indicating that the model offers a physiologically plausible framework for how vibrissal drive and optogenetic excitation of sensory cortex generate perceived duration. The scatter plots of Fig. 4b (inset) show the optogenetic excitation-induced bias in perception, in behavioral data and modelled neurometric output.

The model predicts that when the vS1 firing evoked by the vibration's duration and intensity is integrated, the two stimulus features will be either congruent or incongruent in their contribution to perceived duration. To test this, using the stimulus generalization matrix given in Supplementary Fig. 1, we identified two groups of trials, congruent and incongruent. For duration rats, trials designated as congruent were characterized by stimulus 2 of short duration (264 ms) and low intensity (34, 42, or 52 mm/s) or else long duration (422 ms) and high intensity (78, 96, or 119 mm/s); incongruent trials were characterized by stimulus 2 of short duration and high intensity or else long duration and low intensity (Supplementary Fig. 14). All of these instances of stimulus 2 were judged by the rats in comparison to stimulus 1, which had intermediate duration (334 ms) and intermediate intensity (64 mm/s). The prediction is that performance for congruent trials will be better than performance for incongruent trials. This is because the congruence of intensity with duration (where intensity causes short stimuli to feel shorter or causes long stimuli to feel longer) will lead the perceived duration to be more distant from that of stimulus 1, and will thus make the judgment easier and more accurate. By contrast, the incongruence of intensity with duration (where intensity causes short stimuli to feel longer or causes long stimuli to feel shorter) will lead the perceived duration to be closer to that of stimulus 1, and will thus make the judgment more difficult and less accurate. Accuracy was 74.1% (STD: +/− 1.26) for congruent trials, significantly better ($p < 0.001$) than the 64.9% (STD: +/− 0.76) accuracy for incongruent trials. This analysis substantiates the model's prediction that the intensity feature can act congruently or incongruently with the duration feature.

Although the present study focuses on duration perception, we verified the analogous effect in intensity rats (Supplementary Fig. 14). This shows that, in general, the congruence/incongruence of the irrelevant feature acts on both percepts, supporting the framework of multiplexed coding of distinct percepts. An additional treatment of the duration/intensity confound in intensity rats can be found in[13,20].

The distance between the black and light blue/red curves (Fig. 2b) allows estimation of the direct perceptual effect of vS1 optogenetic intervention (Fig. 4c). A vibrissal stimulus of actual duration 334 ms, absent any direct intervention in sensory cortex, will be perceived as having a veridical duration of 334 ms (black bar). That same stimulus, when accompanied by vS1 photoexcitation (blue light) will be perceived (on average) as having a duration of 372 ms, an optogenetic-derived perceptual dilation of 39 ms. In the control condition, photoinhibition, that stimulus will be perceived (on average) as having a duration of 316 ms, a perceptual compression of 18 ms. These perceptual effects are denoted by the three hourglasses, which accumulate sensory inputs, and thus fill up to a level that is dependent on both the amplitude and the duration of vS1 firing. The empirical observations, coupled with the physiological model for vS1 and downstream integration, offer a detailed picture for how the judgment of the passage of time embodies sensory coding in the cortex.

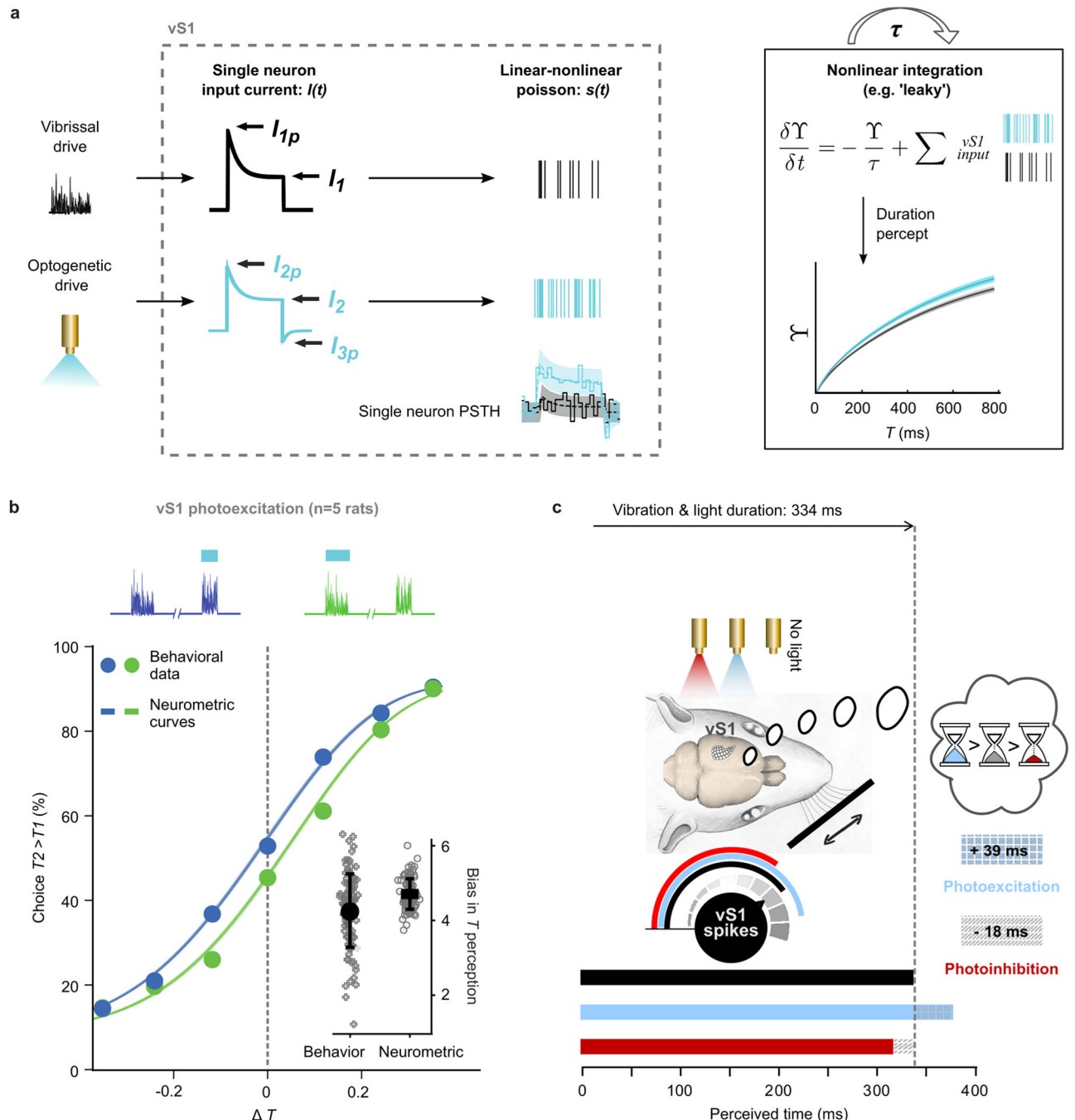

**Fig. 4 | Model of non-linear integration to generate time percept. a** Left box: stimulus-dependent input currents that follow characteristic dynamics for vibrissal and optogenetic drive, converge on vS1 neurons, giving rise to spike trains with Poisson statistics. Example vS1 PSTH is shown in the lower right corner; binned neuronal activity (solid line) in response to a vibrissal stimulus (black), as well as with photoexcitation (light blue), are simulated (dashed line with confidence intervals) by fitting the respective input currents and feeding them to an I/F. Current and I/F parameters in Methods. We simulated a 5000-neuron population based on the distribution of fitting parameters of the entire population of recorded neurons. Right box: the LI receives input spike trains from the simulated vS1 population under conditions including vibrissal (black spike train) and vibrissal plus photoexcitation (blue spike train). The integrator's accumulated quantity is governed by the differential equation. Reading out the generated vS1 neuronal population activity with this LI, we can predict the perceptual shift created by optogenetically increasing firing rate in vS1. **b** Psychometric curves of the neurometric model (points). Inset: comparison of the perceptual shifts between behavioral data and generated neurometric curves for resampled (100×) neuronal and behavioral data. Bias quantified as the difference in percent of choices $T2 > T1$ averaged across all data points, excluding $\Delta T$'s of 0.35 and −0.35 (for details, see Methods). **c** vS1 role in compressing or dilating perceived time by its sensory drive; optogenetic manipulation slows or speeds the accumulation of drive in the perceptual hourglass. The selected changes in perceived time derived from the shift in the point of subjective equality (PSE) in the averaged behavioral data. Source data are provided as a Source Data file.

## Discussion

Applying targeted and controlled optogenetic manipulation of vS1 in one set of rats performing tactile duration discrimination and in another set of rats performing tactile intensity discrimination, this study reveals parallel generation of two percepts, with primary sensory cortex common to both networks. While different mechanisms may be at work in processing the empty interval between two discrete events[6,30–33], time perception accompanying an ongoing stimulus

stream appears to arise within the sensory representation of touch, where it is multiplexed with the neuronal coding of tactile features. The contribution of the coding of stimulus features to sense of time distances our findings from dedicated pacemaker-accumulator operation models[9,11,12], where the accumulator receives onset/offset triggers but is insensitive to the neuronal coding of the stimulus[34]. In existing models where the time percept is constructed from sensory drive[13,18,35,36], no causal link between sensory cortex and the final percept has yet been established. The link established here by means of optogenetic stimulation and multi-channel neuronal recording was not based on degradation of performance as a consequence of interference with neuronal activity[37], as is common in the literature[22,38,39]; indeed, rats performed the task equally as well, as measured by psychometric curve slope, on trials with and without optogenetic intervention (Supplementary Fig. 2b, c). Rather, the evidence of causality is a statistically significant shift - either towards greater perceived duration (consequent to photoexcitation) or lesser perceived duration (photoinhibition) - while the altered percept remained reliable (see also psychometric curve fitting parameters, Supplementary Fig. 4). An explanation for this modulation of the otherwise normal percept derives directly from the computational model, which shows how the measured optogenetically-evoked change in firing would be expected to lead to precisely the observed change in percept.

Embodied within a network extending to the sensory cortex, time perception now becomes amenable to the tools previously restricted to quantifying the representation of stimulus features – tools such as spike counts, firing rates and temporal patterns[19–21,23–25]. In short, the door is now open to unraveling time with the toolbox of sensory coding. One immediate insight from this treatment is that processing likely involves integration of sensory cortical input with long integration time constants, as are found in frontal cortical regions[20].

In our hands, photoexcitation-evoked changes in the vS1 response to vibrissal vibration were seen in the peak of the initial response and, to a lesser extent, in the steady-state response (Fig. 3c). This suggests that photoexcitation acts most prominently at response onset, when stimulus-evoked responses are believed to propagate through local cortical circuits before strong feedforward inhibition clamps down on the excitability of sensory-coding neurons[40]. The early response of sensory cortex appears to comprise part of the stream that is accumulated at successive stages of processing.

Stimulus 1 had a less prominent effect on choice than did stimulus 2. The order-dependent effects parallel the relative psychometric weights that subjects, humans and rats, typically give to the first and second stimulus[26,41–44]. The lower stimulus 1 weight may be an outcome of the formation of a less acute percept due to lower attention, of contraction of the remembered percept towards a long-term prior, of the attribution of stimulus 1 to a prior which acts as the criterion to which stimulus 2 is compared, or some combination of all of the above[41,42,45].

A crucial component of the present model, the accumulation of sensory drive by a downstream integrator, appears to apply to humans[13,17,36,46]. The generality of the model raises the prospect that anomalous sensory coding mechanisms may be one contributing factor in the time misperception at the core of multiple psychiatric disorders[47–49].

## Methods

### Rat subjects

All protocols conformed to international norms and were approved by the Ethics Committee of SISSA and by the Italian Health Ministry (license numbers 569/2015-PR and 570/2015-PR). For the working memory behavior, 20 male Wistar rats (Harlan Laboratories, San Pietro Al Natisone) were caged in pairs and maintained on a 14/10-h light/dark cycle. They were trained and handled on a daily basis and provided with daily environmental and social enrichment. To promote

motivation in the behavioral task, rats were water-restricted for approximately 20 h prior to training or testing sessions; access to food in the cage was continuous. They were tested each weekday in sessions of about 1 h. Part of the data that was analyzed and presented here (rat psychophysical results; Fig. 1b–d, 3 duration and 11 intensity rats) overlaps data used in a previously published paper[13]. Figures 1, 2 overlap in 3 duration rats. The control condition presented in Fig. 2d, was applied in all duration rats following the optogenetic manipulation sessions (Fig. 2b-c), apart from one rat that died before controls were concluded. Neurons presented in Figs. 3a–c, e–i, 4 were recorded from all duration rats with vS1 photoexcitation ($n = 5$); all intensity rats with vS1 photoexcitation ($n = 3$) were included in Fig. 2e and all such rats contributed neurons to Fig. 3d and Supplementary Figs. 8–10.

For the reference memory task, 4 additional rats were trained and tested, following a regime parallel to that described above.

### Behavioral tasks

In the working memory task, to initiate a trial, the rat entered the nose poke, placing its whiskers in contact with a plate connected to a shaker motor (type 4808; Brüel & Kjær[26]). After a pre-stimulus delay of 500 ms, it then received vibrissal stimulus 1 and stimulus 2, separated by a delay of 2000 ms. Then, following a variable post-stimulus 2 delay of 500–750 ms, the auditory "go" cue sounded and the rat had to withdraw and select the left or right spout; reward location depended on either the relationship between the two vibration durations or the two vibration intensities (see below). If the relevant feature was equal for the two stimuli, the choice was randomly rewarded at the left or right spout. The association between stimulus relationship and reward location was randomly assigned to each rat. Early withdrawals aborted the trial and no reward was released. Incorrect choices were followed by a time-out delay of 1–3 s.

Stimuli were noisy vibrations, constructed by stringing together over time a sequence of plate velocity values (motion along the axis of the rod connecting the plate to the motor). Velocities were sampled (1 kHz) from a Gaussian distribution with 0 mean and standard deviation ranging from 25 to 148 mm/s. The speed distribution (absolute values of velocity) was a half-normal (folded) distribution whose mean was equivalent to the standard deviation of the underlying Gaussian multiplied by $\sqrt{(2/\pi)}$. We refer to mean speed as intensity ($I$). A total of 50 unique stimulus traces, "seeds" were created by resampling a given Gaussian; such stimuli differ in local features but are characterized globally by nearly equivalent $I$ (see[20]). When neuronal data are sorted according to $I$, such data are collected from multiple seeds.

Vibration duration was denoted $T$. Durations varied from 112 to 1000 ms (see Supplementary Fig. 1). The differences between the two stimuli making up one trial are expressed by two indices, normalized intensity difference ($\Delta I$) and normalized time difference ($\Delta T$):

$$\Delta I = \frac{I2 - I1}{I2 + I1} \qquad (1)$$

$$\Delta T = \frac{T2 - T1}{T2 + T1} \qquad (2)$$

where $I1$ and $I2$ are the intensities, and $T1$ and $T2$ are the durations of stimuli 1 and 2, respectively. The values of intensities and durations applied are spaced logarithmically and are presented in Supplementary Fig. 1.

Duration rats were trained and tested using a rule where reward location, left or right spout, was determined by the sign of $\Delta T$, with $\Delta I$ irrelevant. Intensity rats were trained and tested using a rule where reward location was determined by the sign of $\Delta I$, with $\Delta T$ irrelevant.

In test sessions, each stimulus combination assumed $T$ and $I$ values from the stimulus generalization matrix (Supplementary Fig. 1). Thus, rats received a random combination of 10 intensity pairs ($I1$, $I2$) x

10 duration pairs ($T1$, $T2$) during each training session. Moreover, in each trial, the relevant and irrelevant features could be congruent ($I2 > I1, T2 > T1$ or $I1 > I2, T1 > T2$) or incongruent ($I2 < I1, T2 > T1$ or $I1 < I2, T1 > T2$). This randomness in congruence required the rat to act upon only the relevant feature in order to perform above chance. To obtain psychometric curves for duration, $T1$ took a fixed value of 334 ms while $T2$ spanned seven possible durations, giving a range of $\Delta T$ from −0.35 to 0.35. The same design was used to obtain psychometric curves for intensity, with $I1$ fixed at 65 mm/s, while $I2$ spanned seven possible values; $\Delta I$ ranged from −0.3 to 0.3.

In the reference memory task, like in the working memory task, the rat initiated a trial by entering the nose poke. After a pre-stimulus delay of 500 ms, vibrissal vibration was initiated, with $I = 63$ mm/s. Seven possible stimulus durations ($T$) were applied; three belonged to the short category (200, 300, 400 ms) and three to the long (600, 700, 800 ms). Following a variable post-stimulus delay of 500–750 ms, the auditory "go" cue sounded and the rat had to withdraw and select the left or right spout; reward location depended on stimulus duration category; the intermediate stimulus duration (500 ms) was randomly rewarded either as short or long. Two of the rats were assigned to the long stimulus/turn right rule, the other half long/left. Early withdrawals aborted the trial and no reward was released. Incorrect choices were followed by a time-out delay of 1–3 s.

## Targeted virus injections in vS1

After rats reached stable behavioral performance in their designated task, they were anesthetized with 2–2.5% Isoflurane in 100% oxygen delivered through a customized plastic snout mask. Target regions were accessed by craniotomy, using standard stereotaxic technique. The vasculature visible on the brain surface was used as a reference for cortical maps[50]. Photos of the brain surface of vS1 were made with a 5× Zeiss microscope connected to a webcam and further used to document electrode insertion and injection sites. Single tungsten electrodes (100–500 kΩ impedance; FHC) were inserted to a depth of ~750 µm and the whisker constituting the neuronal population's strongest input was assessed by stimulation with a hand-held probe. Neuronal populations with receptive fields on whisker rows C-D and columns 4–6 were targeted. AAV5-CaMKIIa-hChR2(H134R)-EYFP or AAV5-CaMKIIa-eNpHR3.0-EYFP (UNC vector core) was prepared by standard procedures[51]. Reference memory rats received only the AAV5-CaMKIIa-hChR2(H134R)-EYFP treatment. A 10 µl Hamilton syringe was filled with 4–6 µl of virus solution (stained with Fast Green FCF). Injections of 0.5 µl of virus solution were made at depths of 800 and 1600 µm in 3–4 barrel-columns with identified receptive fields. The skull opening was conserved with custom-made cylindrical implanted cranial windows. Four to six screws were fixed in the skull as support for dental cement. Two screws served as reference and ground and were connected via a silver wire to a 2-pin connector that was embedded in dental cement. At the conclusion of the operation, rats were treated with antibiotic (Baytril; 5 mg/kg; i.p.), analgesic (Rimadyl; 2.5 mg/kg, i.m.), atropine (ATI; 2 mg/kg, s.c.) and with sterile saline to rehydrate (5 ml, s.c.). A local antibiotic ointment was applied around the cutaneous wound to improve the healing. Tissue was washed regularly and treated with antibiotics in the weeks after surgery through the cylindrical implanted cranial windows. During the recovery period, rats had unlimited access to water and food.

## Implantation of opto-electric microdrive

In a second operation 3–4 weeks after virus injection, the opto-electric microdrive was implanted through the cranial window. Injection sites were identified by vasculature landmarks and mapping from the previous surgery. For chronic electrophysiological recording concomitant with optogenetic stimulation, we collaborated with CyNexo to design an opto-electric microdrive (aoDrive, https://www.cynexo.com/portfolio/neural-drives/, see also Fig. 2a, middle). Each drive incorporates up to 15 single FHC tungsten electrodes and an optic fiber (Ø: 230 µm, NA: 0.67, Plexon). Electrodes and fiber can be independently moved in depth with a total range of 4.5 mm. Electrodes were lowered until neuronal responses to light delivery (PlexBright LED, Blue: 465 nm, or Orange: 620 nm for inhibition, Plexon) were observed. Subsequently, microdrives and TDT connectors were embedded in dental cement. The opto-electric microdrive provided signals for 3–6 months after the surgery. Electrophysiological recording and optogenetic intervention in the behaving animal began 7–10 days following the second implantation surgery. Rats, whether in the working memory or reference memory task, performed light-on trials with no signs of surprise or disorientation from the very first trials of the first session in which light-inputs were applied.

## vS1 recordings and optogenetic behavioral experiments

Extracellular activity was pre-amplified, filtered and digitized using the digital TDT recording system (Tucker David Technologies) along with task-relevant data, such as position sensors and light/motor stimulation signals to synchronize external events with physiological recordings. Recording depths varied between 700–1500 µm, to cover layers IV and V. Signals were sorted into single and multi-unit neuronal clusters, as verified through standard indices using UltraMegaSort2000[52] and both single and multi-unit clusters were used for all further analysis. The headstage and optic fiber patch cables (custom made, Ø: 230 µm, NA: 0.67) were connected to the implant and the cables were held by a rubber band to limit weight on the implant. Light output intensity (10–12 mW) from the tip of the patch cable was measured (Thorlabs) weekly to ensure stable optogenetic excitation/inhibition effect. Light delivery (465 nm) for the optogenetic excitation and external light experiment was synchronized with the onset of vibrissal stimulation. Offset time of light delivery and vibrissal stimulation was identical. Light delivery (620 nm) for the optogenetic inhibition experiment was slightly adapted in order to account for the biophysical mechanisms of eNpHR3.0[53]: (1) light was initiated 50 ms preceding onset of vibrissal stimulation to reach an effective hyperpolarization, accounting for the slower time constants of eNpHR3.0 as compared to Chr2(H1340). (2) Light was dimmed with an offset ramp, starting 200 ms before and ending at 0 ms with respect to termination of vibrissal stimulation; this protocol minimized rebound activation.

In the reference memory task, rats establish their internal decision criterion within each experimental session based on a weighted average of the perceived stimuli in the preceding trials[45]. Therefore, recording sessions were started with a criterion-setting period of 100 trials without optogenetic stimulation. In the subsequent optogenetic stimulation portion of the session, 50% of trials (designated randomly) consisted of photoexcitation during vibration, while the remaining 50% of trials were control trials with vibration only. In the photoexcitation trials, blue light (460 nm) was delivered to the left vS1 (injected with Chr2(H1340)), starting 25 ms after vibration onset and terminating 25 ms before vibration offset. To verify that effects did not depend on a continuous light source, stochastic light amplitude was applied. Stochastic light signals were constructed by stringing together over time at a rate of 40,000 samples/s, a sequence of voltage values sampled from a Gaussian distribution with 2.5 V mean and 2 V standard deviation. The voltage values were fed as analog input to the LED (aoLED, Cynexo: https://www.cynexo.com/portfolio/aoled-optogenetics/), whose output was 12–13 mW when measured with a constant drive of 5 V. To avoid retinal activation during optogenetic light delivery, external stochastic light (460 nm) was delivered in the otherwise dark experimental box throughout the entire recording sessions.

## Histological examination

At the conclusion of the study, electrolytic lesions were made around the tips of the electrodes to mark the recording sites. To identify

Opsin-expressing neurons, we counterstained with blue-fluorescent Nissl stain (NeuroTrace 435/455, ThermoFisher) to visualize cortical layers, and performed antibody staining (Primary antibody: Anti-VGLUT2, Synaptic Systems, 1:750 dilution; Secondary antibody: Alexa Fluor™ 594, Thermo Fisher Scientific, 1:500 dilution) to discern the barrels in layer IV. Whole coronal slice (50 μm thickness) images were taken with confocal microscope (4x, Nikon).

## Analysis of behavioral data

To generate duration psychometric curves, we used trials in which $T1$ was 334 ms while $T2$ ranged from 161 ms to 694 ms. For intensity psychometric curves, we used trials in which $I1$ was 64 mm/s while $I2$ ranged from 34 mm/s to 119 mm/s (Supplementary Fig. 1). The rat's choice (proportion of trials in which the stimulus 2 was judged as more intense or longer in duration than stimulus 1) for each stimulus pair was then plotted. A four-parameter logistic function was fit to the psychometric data using the nonlinear least-squares fit in MATLAB (MathWorks, Natick, MA), as follows. The psychometric curve for duration was given by

$$P(T2 > T1) = \gamma + (1 - \lambda - \gamma)\frac{1}{1 + \exp(-(\Delta T - u_T)/v)} \quad (3)$$

and for intensity the curve was given by

$$P(I2 > I1) = \gamma + (1 - \lambda - \gamma)\frac{1}{1 + \exp(-(\Delta I - u_I)/v)} \quad (4)$$

where $\Delta T$ and $\Delta I$ are the normalized stimulus differences, $\gamma$ is the lower asymptote, $\lambda$ is the upper asymptote, $1/v$ is the maximum slope of the curve and $u_T$ and $u_I$ are $\Delta T$ and $\Delta I$ at the curve's inflection point, for the duration and intensity curves, respectively. Since observed experimental data are expressed as choice (%), the proportion of trials in Eqs. (3–4) was multiplied by 100 for purposes of illustration.

The raw data (choices for each value of $\Delta T$ and $\Delta I$) as well as the fit parameters were then used as measures of acuity and bias. Bias in perceived duration was measured by sorting the trials according to $I$ and then, with the $T$ values pooled, averaging the percent judged $T2 > T1$ for each value of $\Delta I$. This isolates the effect of $\Delta I$ on perceived duration. For illustration (Fig. 1d), the above bias measure was normalized by subtracting the choice data for $\Delta I = 0$ (where $I$ does not exert a bias). The bias in perceived duration caused by changes in the irrelevant feature ($\Delta I$) was negligible when the relevant feature difference ($\Delta T$) was 0.35 or −0.35 because even the presence of an intensity bias did not cause a shift in choice; such "easy" trials were excluded from the bias analysis. Likewise, the change in the percent of trials judged $I2 > I1$ caused by changes in $\Delta T$ was negligible when the relevant feature difference ($\Delta I$) was 0.3 or −0.3.

## Resampling and bias statistics

To quantify the effect of optogenetic manipulation of vS1 on duration and intensity perception, we resampled the original data set to create 1,000 sets of statistically comparable data, with the same trial size as the original data, allowing for replacement. Each set of resampled responses was parametrized by fitting the logistic function of Eq. (3). A support vector machine (SVM) classifier (MATLAB, fitcsvm function) quantified the linear separation between data points (PSE, acquired from the fitted curve versus averaged percentage of choice of the resampled data, not including fitting) with and without optogenetic intervention, making use of 10-fold cross validation to measure the classification error. Specifically, the data were partitioned into 10 random sets. Then, 9 of these were used to train an SVM classifier and the remaining set served as a test. This procedure was repeated 10 times and the statistics for each repetition were combined, giving the rightmost plots of Fig. 2b–d. Statistical significance of the bias in

duration and intensity judgements between the different condition pairs (Fig. 2b-e, middle, Supplementary Fig. 2a, 5) was tested by subtracting at each resampling iteration the averaged percentage of choice (excluding easy trials) between the respective condition pair [bias$_{it}$ = mean(% choice$_{stim2>stim1}$ condition 1) − mean(% choice$_{stim2>stim1}$ condition 2)]. $p$-value was estimated by (# of iterations with bias >= 0) / 1000. To uncover optogenetic effects when not overridden by strong sensory evidence, statistical significance in Supplementary Fig. 2a (middle panel) was tested on the stimulus pairs where $\Delta T = 0$ (for duration rats).

## Neuronal data

Spike trains were aligned to the stimulus onset or else offset, depending on the aim of the analysis. In Fig. 3b, individual neurons' response was generated by plotting the average firing rate over trials, shifting in 1 ms steps. To reduce the effect of noisy fluctuations, a centered 40 ms sliding window was used. Firing rate was then z score–transformed by subtracting each neuron's spontaneous activity rate (measured from 800 ms before stimulus onset up to the stimulus onset) from response rate per time bin during stimulus presentation. The outcome was divided by spontaneous activity variance.

Population PSTHs (Fig. 3c) were generated by plotting the average population response shifting in 1 ms steps for each stimulus duration. To avoid forward leakage across stimulus onset/offset boundaries the PSTH values were derived by convolution of the averaged data with a half-Gaussian ($\sigma = 25$ ms). The half-gaussian was "flipped" on either side of a boundary and in the middle of the PSTH inasmuch as the first portion of the PSTH was formed with the half-gaussian tail oriented forward while the second portion was formed with the tail oriented backward.

Intensity coding was tested from vS1 neurons ($n = 138$), recorded in EYFP-ChR2(H134R)-expressing rats ($n = 3$) performing the intensity task. The criterion to classify individual neurons as coding ($n = 39$) or non-coding ($n = 99$) was a significant difference in the linear correlation between firing rate and stimulus intensity. The observed correlation coefficient was then compared to the distribution of linear correlation between the firing rate and the shuffled stimulus intensities labels (iterated 1000× trials, resampling method, $p < 0.05$). For quantifying intensity coding of the vS1 neuronal population and the effects of optogenetic excitation to intensity coding (Fig. 3d and Supplementary Fig. 10), data resampling was performed as suggested earlier[13] on a subset of neurons (90% coding, 10% non-coding).

Response onset and offset times (Figs. 3h, i) were registered when the population mean firing rate underwent an upwards and downwards (respectively) crossing of the threshold set at 2.2× STD of the smoothed firing rate obtained 480 to 20 ms before stimulus onset. Outliers, defined as onset/offset times exceeding 25 ms from stimulus onset/offset, were removed.

## Model for vS1 activity

The analysis was built on the spiking activity of 240 vS1 units recorded in 5 rats over the full range of vibrissal stimulation and optogenetic excitation conditions. To permit more robust neurometric measures, we modeled a larger data set replicating the properties of actual recordings by means of a Gaussian Mixture Model (GMM)[29]. The key observations justifying this form of model are that functional properties are diverse across neurons and the responses of single neurons are variable across trials. The methodology is presented in two steps: (1) constructing a parametric model for single neuron variability and fitting it to each recorded unit, and (2) Gaussian mixture model to reproduce vS1 population diversity.

(1)   The parametric model for neuron variability is based on the following observations: (i) the response of a neuron varied across trials notwithstanding constant stimulus conditions, (ii) in the absence of stimulation, vS1 neurons showed ongoing activity, (iii)

when the vibrissal stimulus was presented, many vS1 neurons rapidly increased their firing rate and then adapted to steady state, (iv) simultaneous optogenetic and vibrissal stimulation commonly led to a higher firing rate but did not alter the temporal profile of response. However, firing rate sometimes dropped after the light was turned off a phenomenon sometimes referred to as post-stimulation suppression[54].

Given the recorded units' observed spike variability, we assume that the activity of the $i$th unit, $k^{(i)}$, in time bin $t$, follows a Poisson distribution.

$$P(t) = \frac{f_r^{(i)}(t)^{k_i} e^{-f_r^{(i)}(t)}}{k!} \qquad (5)$$

where $f_r^{(i)}(t)$ is the unit's firing rate. We write a parametric model for the underlying rate of the Poisson distribution and infer its parameter values based on the measured PSTH of each recorded unit. This spike generation model assumes that the $i$th unit receives a constant background current $I_0^{(i)}$. We assume that when the vibrissal stimulus is turned on, it elicits a mechanoreceptor-derived current of the following form

$$I_M^{(i)}(t) = \left[ I_1^{(i)} + \left( I_{1p}^{(i)} - I_1^{(i)} \right) e^{-\frac{t-t_{0M}^{(i)}}{\tau_M^{(i)}}} \right] W\left(t, T, t_{0M}^{(i)}\right) \qquad (6)$$

where $I_{1p}^{(i)}$ is the peak input current that follows immediately after stimulus onset, $I_1^{(i)}$ is the steady state current, $t_{0M}^{(i)}$ is the onset of the mechanoreceptor-elicited current, and $\tau_M^{(i)}$ is the decay time constant from $I_{1p}^{(i)}$ to $I_1^{(i)}$. $W(t, T, t_{0M}^{(i)})$ is a window function that determines when the stimulus evokes a current that is non-zero:

$$W\left(t, T, t_{0M}^{(i)}\right) = expit\left(t - t_{0M}^{(i)}\right) expit\left(-\left(t - T - t_{0M}^{(i)}\right)\right) \qquad (7)$$

where

$$expit(x) = \frac{1}{(1 + \exp(-x))}.$$

When the optogenetic excitation is turned on, it elicits a current of the following form:

$$\begin{aligned} I_O^{(i)}(t) = &\left[ I_2^{(i)} + \left( I_{2p}^{(i)} - I_2^{(i)} \right) e^{-\frac{t-t_{0o}^{(i)}}{\tau_O^{(i)}}} \right] W\left(t, T, t_{0o}^{(i)}\right) \\ &+ I_{3p}^{(i)} e^{-\frac{t-T-t_{0o}^{(i)}}{\tau_I^{(i)}}} expit\left(t - T - t_{0o}^{(i)}\right) \end{aligned} \qquad (8)$$

where $I_{2p}^{(i)}$ is the peak input current that follows immediately after light onset, $I_2^{(i)}$ is the steady state current, $t_{0o}^{(i)}$ is the time of onset of the current, $\tau_I^{(i)}$ is the decay time constant from $I_{2p}^{(i)}$ to $I_2^{(i)}$, and $I_{3p}^{(i)}$ is the activation current that follows light offset. $W(t, T, t_{0o}^{(i)})$ is the same windowing function as in the mechanically elicited current.

The total input current received by the $i$th unit is

$$I^{(i)}(t) = I_0^{(i)} + I_M^{(i)}(t) + I_O^{(i)}(t) \qquad (9)$$

Input current to output firing rate curve (I/F curve) is modeled as a generalized sigmoid[55]:

$$f_r^{(i)}(t) = \lambda^{(i)} \left[ 1 - \left( \frac{1}{1 + \exp\left(10 I^{(i)}(t)\right)} \right)^{\frac{1}{v^{(i)}}} \right] + 10^{-4} \qquad (10)$$

where $\lambda^{(i)}$ denotes the maximum firing rate and $v^{(i)}$ represents the non-linear scale of the generalized sigmoid curve. The current $I^{(i)}(t)$ is multiplied by a constant scaling factor 10, to improve the fit stability due to the different scales of the many parameters involved. The added $10^{-4}$ constant helps to stabilize the fit in very low-firing neurons.

We used the pymc3 package[56] to infer the 12 parameters given the observed PSTH of the neurons in 10 ms wide time bins. The model output accurately fitted the mean firing rate and the variability in the spike trains for the 240 recorded units.

(2) The Gaussian mixture model exploits the diverse properties of individual vS1 neurons to make estimates of large populations on the basis of limited quantities of recordings. The reasoning is that a recorded neuron, unit $i$, is a randomly sampled member of a broader group of vS1 neurons with similar properties (i.e., similar parameters according to the single neuron variability model described above). We label this group $g_i$. Unit $i$ emits spikes with a Poisson probability distribution (Eq. (10)).

In detail, we first assume that vS1 is made up by a mixture of $G$ qualitatively different classes of neurons, where classes are defined by the parameter values of the single neuron model. Each class can be represented as a Multivariate Gaussian in a subspace $A$ of the model's parameter space, where A is determined by $I_0$, $I_1$, $I_{1p}$, $I_2$, $I_{2p}$, $I_{3p}$, $\lambda$ and $v$. The parameters, $\tau_M$, $\tau_O$ and $\tau_I$ showed only minimal variations among neurons, making it likely that they are population-specific and not neuron-specific; thus we chose to fix their values to the median of the fit for all neurons: $\tau_M = 48$ ms, $\tau_O = 49$ ms and $\tau_I = 28$ ms. The parameters $t_{0M}$ and $t_{0o}$ are defined by the experimenter and therefore kept constant.

Class membership is derived from a Dirichlet process[57]. The full generative process underlying real neuronal data can be written as

$$\begin{aligned} N\left(\overrightarrow{\mu_{gi}}, \Sigma_{gi}\right) &\sim DP(N, \alpha) \\ \overrightarrow{a_i} &\sim N\left(\overrightarrow{\mu_{gi}}, \Sigma_{gi}\right) \\ P(\overrightarrow{a_i}, t) &\sim Poisson(f_r(\overrightarrow{a_i}, t)) \end{aligned} \qquad (11)$$

where the $i$ subindex corresponds to the recorded unit, the $g_i$ subindex represents the group to which the $i$th unit belongs to, $\overrightarrow{a_i} \in A$ is the vector of parameters $I_0^{(i)}$, $I_1^{(i)}$, $I_{1p}^{(i)}$, $I_2^{(i)}$, $I_{2p}^{(i)}$, $I_{3p}^{(i)}$, $\lambda^{(i)}$ and $v^{(i)}$, $DP$ is a Dirichlet process, $\alpha$ is the concentration parameter, and $\overrightarrow{\mu_{gi}}$ and $\Sigma_{gi}$ are, respectively the $g_i$th class mean and covariance.

We infer the mixture weights, $\overrightarrow{\mu_g}$ and $\Sigma_g$ by using the expected values of the independently inferred parameters from the single neuron variability model as the observed $\overrightarrow{a_i}$ vectors for each unit. We then used scikit-learn[58] built-in `BayesianMixtureModel` class to infer the suitable class proportions, and each class's $\overrightarrow{\mu_g}$ and $\Sigma_g$. We set the $\alpha$ concentration hyperprior to $10^{-6}$ to favor assigning significant weight to a larger number of classes, but also set $G = 5$ to prevent excessive granularity.

This generative process and the Bayesian Mixture Model allow us to model a 5000-unit vS1 population. The resulting spike trains are statistically consistent with observed single neuron variability and diversity in vibrissal stimulation and optogenetic excitation response. We take the full set of modeled neurons to be the drive $f_{vS1}(t)$ for any given stimulation condition, as

$$f_{vS1}(t) = \sum_j k^{(j)} | t \qquad (12)$$

## Model for perceived stimulus duration

The perceived stimulus duration, $\Upsilon$, is modeled by the leaky integrator differential equation

$$\frac{d\Upsilon}{dt} = -\frac{\Upsilon}{\tau} + f(t) \qquad (13)$$

where $\tau$ is the leaky integrator time constant and $f(t)$ is the external drive. The drive is written as

$$f(t) = f_{vS1}(t) + \xi(t) \qquad (14)$$

where $f_{vS1}(t)$ is neuronal activity in vS1, including vibrissal- or optogenetic -stimulation evoked responses and $\xi(t)$ is ongoing firing unrelated to vibrissal or optogenetic stimulation. We approximate $\xi$ as a Gaussian stochastic variable with mean $\mu_b$ and variance $\sigma^2_b$.

Leaky integration is not specified as a unique physiological process; rather, it represents the dynamics governing the percept ($\Upsilon$) in a manner that quantitatively accounts for the rat's judgments as a function of combined vibrissal stimulation and optogenetic excitation evoked responses.

## Model for choice

Given the leaky integrator dynamics of $\Upsilon(t)$, the model for $f(t)$, and the approximation of the Poisson variability in $f_{vS1}(t)$, we can solve the stochastic differential Eq. (13) as shown previously[13,59]. $\Upsilon(t)$ follows a Gaussian distribution and its expected value and variance are equal to

$$E[\Upsilon(t)] = E[\Upsilon(0)]e^{-\frac{t}{\tau}} + \mu_b\left(1 - e^{-\frac{t}{\tau}}\right) + \int_0^t e^{-\frac{t-t'}{\tau}} f_{vS1}(t) dt' \qquad (15)$$

$$Var[\Upsilon(t)] = Var[\Upsilon(0)]e^{-\frac{2t}{\tau}} + \sigma^2_b\left(1 - e^{-\frac{2t}{\tau}}\right) + \int_0^t e^{-\frac{2(t-t')}{\tau}} f_{vS1}(t) dt' \qquad (16)$$

During stimulus delivery, the rat's percept of elapsed time evolves through Eq. (13). The percept of total stimulus duration is given by $\Upsilon$ at the time of stimulus offset ($\Upsilon(T)$). We then compute the probability distribution for each stimulus duration, $\Upsilon(T1)$ and $\Upsilon(T2)$, in the delayed comparison task. This gives the probability that $\Upsilon(T2)$ is greater than $\Upsilon(T1)$ as

$$P(\Upsilon(T2) > \Upsilon(T1)) = \frac{1}{2} + \frac{1}{2} erf(d') \qquad (17)$$

where

$$d' = \frac{E[\Upsilon(T2)] - E[\Upsilon(T1)]}{\sqrt{2(Var[\Upsilon(T2)] + Var[\Upsilon(T1)])}} \qquad (18)$$

We can assume two types of trials[60] – those in which the rat encoded $\Upsilon(T2) > \Upsilon(T1)$ and used the two representations to make a choice ("attended" trials) and those in which choice was unrelated to the evoked sensory representations ("lapse" trials). We assume that in attended trials the rat judged $T2 > T1$ whenever $\Upsilon(T2) > \Upsilon(T1)$; in lapse trials, the rat chose at random according to some choice bias probability $b_L$.

The probability that the rat judged $T2 > T1$ is then

$$P(T2 > T1) = p_L b_L + (1 - p_L)\left[\frac{1}{2} + \frac{1}{2} erf(d')\right] \qquad (19)$$

where $p_L$ is the probability of a lapse trial.

## Fit of the behavioral psychometric data

We constructed 100 independent population proxies for vS1 firing, each made up of 5,000 neurons using the model described in (10). We then fit a common $\tau$, $\mu_b$, $\sigma^2_b$, $p_L$ and $b_L$ across all 100 populations using a maximum likelihood estimate based on Eq. (19) with L2 parameter regularization for $\tau$. The weight of the regularization was set to 0.01 and its center was placed at 600 ms. The resulting parameter values are listed in table (Supplementary Table 1).

Using the resulting parameters for the neuronal population proxies, we computed the model's predicted psychometric curves (Fig. 4b), and the behavioral bias (overall predicted probability of choosing $T2 > T1$, Fig. 4b inset).

## Reporting summary

Further information on research design is available in the Nature Portfolio Reporting Summary linked to this article.

## Data availability

The data that support the findings of this study are deposited to Zenodo and can be accessed through: https://doi.org/10.5281/zenodo.10548788[61]. The data sets include rat behavioral data, neuronal data recorded in vS1 of rats that perform vibration intensity and duration discrimination task, as well during optogenetic manipulation. Any additional information will be available from the authors upon request. Source data are provided with this paper.

## Code availability

The code for the different analyses in this study, as well as code used to generate the main plots is deposited to GitHub and can be accessed via https://github.com/arashfassihi/Direct-contribution-of-the-sensory-cortex-to-the-judgment-of-stimulus-duration. Any additional information will be available from the authors upon reasonable request.

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

## Acknowledgements

We thank A. Toso, N. Nikbakht, I. Hachen and J. Nicholls for helpful discussions and valuable insights and F. Manzino (CyNexo) and S. Parusso (CyNexo) for their invaluable technical assistance. A. Toso provided helpful comments on the manuscript. S. Sorella and B. Trovo assisted in animal training and behavioral data acquisition. We kindly acknowledge M. Riggi and M. Grandolfo for helpful comments on histology and imaging. M. Mahn, A. Akrami, J.-J. Sun are kindly acknowledged for helpful comments on optogenetics methodology. This work was supported by Human Frontier Science Program, project RGP0017/2021 (MED), European Research Council advanced grant CONCEPT, project: 294498 (MED), European Union FET grant CORONET, project: 269459 (MED), European Union Horizon 2020 MSCA Programme NeuTouch under Grant Agreement No 813713, Italian Ministry of Education, Universities and Research grant HANDBOT, project: GA 280778 (MED). SR was supported by a NARSAD Young Investigator Grant from the Brain & Behavior Research Foundation (30389).

## Author contributions

Conceptualization: A.F., S.R. and M.E.D. Optogenetic methodology: S.R., M.G. and A.F. Modeling methodology: L.P., S.R., A.F. and M.E.D. Investigation: S.R., A.F., M.R., F.P. and L.P. Formal analysis: A.F., S.R., L.P., M.E.D. Data curation: A.F., S.R., M.R. and M.E.D. Writing -Original Draft: M.E.D., S.R., A.F., L.P. and F.P. Writing – Review & Editing M.E.D., S.R., A.F. Funding Acquisition: M.E.D.

## Competing interests

The authors declare no competing interests.

## Additional information

**Supplementary information** The online version contains Supplementary Material available at https://doi.org/10.1038/s41467-024-45970-0.

