## [Peer Review File · Nature Communications]

REVIEWER COMMENTS

Reviewer #1 (Remarks to the Author):

Combining optogenetic perturbation with a psychophysics discrimination task, Reinartz et al. investigate the contribution of vibrissal somatosensory cortex (vS1) activity to the perception of sensory stimulus duration and intensity. The authors train two groups of rats to either discriminate the duration or the intensity between pairs of vibrissal stimuli. They then use optogenetic photoactivation and photoinhibition of vS1 activity to assess changes in animals' perception of intensity or duration. Through extracellular electrophysiological recordings, the authors then examine neural codes for duration and intensity perception, informing a model to recreate observed psychophysical data. The authors aim to challenge a common belief that "the primary sensory cortex merely relays start and stop signals", and instead propose that "activity of vS1 plays a direct role in time judgement". However, because the optogenetic perturbation conducted in this study covered the entire duration of the stimulus, including the start and stop, the behavioural effects observed cannot rigorously differentiate the above hypotheses. To directly test them, optogenetic manipulation should be restricted to a central portion of the stimulus in which onset/offset triggers are unaffected. This study also demonstrates the potential to discriminate between sensory perception of intensity and duration through vS1 spike rate coding vs spike count coding respectively. However, the analyses shown in the study remain somewhat superficial. We propose further analysis to support this claim in detailed comments below.

Overall, this study provides evidence for the neural correlates of sensory duration perception in vS1 and that vS1 activity may be required for such perception. However, evidence in favour of the proposed models that vS1 activity provides a continuous input to a downstream time perception integrator, in contrast to only providing a stimulus onset and offset signals, are not strongly supported by the experimental data (as discussed above). Furthermore, we were concerned with the strong claims from the optogenetic perturbation experiments with relatively small effect sizes, and at times, missing control experiments. Finally, the wording of the manuscript can be improved for clarity as well as to more accurately reflect the topics investigated (stimulus duration rather than time perception). As a result, despite the potential interest in the underlying topic, we have limited enthusiasm for the publication of this study before these concerns can be addressed.

Major concerns:

1) Design, implementation and interpretation of the optogenetic perturbation experiments

Design: The main conclusion of the paper rests on the optogenetic perturbation experiments. In particular, the authors aim to challenge the hypothesis that "the primary sensory cortex merely relays start and stop signals", and instead propose that "activity of vS1 plays a direct role in time judgement". Although the results shown in the paper are consistent with the latter hypothesis, it is also not inconsistent with the former hypothesis. To directly tease them apart, a key experiment should be optogenetic perturbation where the laser is restricted to a central portion of the stimulus in which

onset/offset triggers are unaffected. We believe that this experiment is needed to support the main conclusion of the manuscript.

Control: To control for the non-neural-activity related effects of laser stimulation (including tissue heating, retinal activation during photo-stimulation etc), eYFP-control experiments should be carried out for photostimulation and photoinhibition experiments.

Presentation and interpretation: In Fig. 2B-D, the authors mainly compared the psychometric curves across different perturbation conditions. What about the effect of each experiment relative to the control condition? “No light” control psychometric curves should also be plotted for each of these panels. This would be especially illustrative for manipulation during stimulus 1, where the effect size is small. For clarity, significance should be determined between these control curves and the experimental curves, alongside significant difference between centre points. It would also be helpful if the authors can quantify the multiple parameters related to the change in the overall psychometric curves (lapse, slope, intercept etc) rather than just focusing on centre points.

Finally, we were quite surprised to see that the authors used constant light delivery at relatively high laser power levels for the ChR2 experiments, which may lead to tissue heating and unphysiological activation of cells. Light power should be reported as a range as opposed to ‘ ≥ 10 mW’, and justification given for the chosen light power and light stimulation profile. Providing pulsed photostimulation at frequencies within the physiological range for vS1 neurons would reduce potential tissue heating and opsin deactivation.

2) More detailed description and quantification of vS1 neural population in light on/off conditions:

As shown in Figure 3A, there are two groups of cells identified in vS1, light- and vibrissal -responsive neurons. Little exploration of the ‘heterogeneous’ recorded vS1 population in EYFP-ChR2(H134R) rats is presented besides 2 example neurons in Fig. 3A. Further analysis of other neuron clusters and their response to photostimulation would be helpful to identify how perturbation affects vS1 population activity. The authors should at least show more individual neuron examples in Supp figures, and provide population summaries beyond just Fig. 3B/C.

The firing rate profile in Fig. 3C of vS1 population activity during vibrissal stimulation shows an initial sharp increase in firing rate followed by a lower sustained firing until stimulus offset. The vibrissal stimulation plus photostimulation condition shows an almost identical pattern with a higher firing rate in the initial peak. This is reported as an evenly distributed boost in the vibrissae-evoked response (L266), but it appears that the increase is largely limited to the first 100 ms. The authors should provide an explanation as to why this initial peak might be the prevailing feature in light-on and light-off conditions and why the primary difference between light-on and light-off conditions is a change in magnitude of the initial peak. Similarly, recording firing rate in fig. 3E during the final 100 ms of stimulus presentation removes information about this initial peak from analysis, despite it being included in the count coding analysis of fig. 3F (we computed counts from vibrissal-stimulus onset to offset (L291)). This choice should be consistent across analysis and maximise information contained within the population activity.

Additionally, there is a sharp decrease in spike rate at the point of stimulus offset in both light-on and light-off conditions that appears to be a break in the traces. Can the authors explain why we are seeing

such a sharp, almost instantaneous decline in activity rather than a gradual decrease of spike rate after light offset?

3) Trial-by-trial neural-behavioural correlation analysis to support the count coding hypothesis

In Fig. 3D-G the authors show evidence that support a rate coding of intensity and a count coding of duration. It would be great if the authors can conduct some trial-by-trial neural-behavioural correlation analysis to further support these mechanisms. Specifically, across trials with the same stimulus duration, does the trial-by-trial fluctuation in spike count correlate with the probability of behavioural report? Similarly, across the trials with the same stimulus duration, does the trial-by-trial fluctuation in spike rate not correlate with choice probability? These analyses can provide further evidence for the count coding of duration in vS1.

4) Wording/writing can be improved for clarity and accuracy:

The suggested title implies that this study gives evidence for a role of the somatosensory cortex in general time perception, which may be misleading. The manuscript focused on studying the perception for the duration of a tactile stimulus, which requires tracking of salient sensory cues. This process may be different from a sensory-modality-independent time perception. Consequently, we suggest that the authors update their title/abstract accordingly and maintain this clear distinction between time perception and sensory input duration throughout the paper.

In general, the manuscript can be improved in terms of the clarity of the writing. For example, many parts of the Results section reads like part of the Figure legend. Rather than describing what is plotted in a figure, it would be helpful for the readers if the authors could instead describe and synthesize the results.

Minor concerns:

1) Only averaged psychometric curves are shown throughout the paper, with only a few showing error bars. Individual psychometric curves should be shown to reveal variance across individual rats. In addition, when the authors quantify the effects of optogenetic perturbations across animals, individual differences should be taken into consideration as well as the average main effects (e.g. using mixed effects in GLME).

2) Fig. 2: What happened after optogenetic inhibition for intensity rats? Have the authors conducted this experiment for comparison?

3) Fig. 2: Only two subjects are used for inhibition in duration rats. Considering that the results are somewhat noisy, it might be important to increase this sample size.

4) Fig. 2E: missing bar for control rats (Fig. 2E right)? It does not seem like that the data is averaging at exactly 0?

5) The high variability in control rats shown in figure 2E (right) makes it unclear whether bias in T perception is actually significantly different between perturbation vs control conditions? This seems inconsistent with the average curve results in 2E middle. Is this because most of the data (trials) come from a few animals?

6) Supp. Fig 2A (middle): External light control rats show a much stronger bias to choosing $T2 > T1$ than would be expected from Fig. 1C. Does this bias remain in control rats when the LED is not present? The psychometric curve of the no light condition for these rats should be plotted alongside the external light curves; and any difference between the external light and no light conditions should be acknowledged.

Reviewer #2 (Remarks to the Author):

Dear All,

I have read with great care the submission by Reinartz et al, and while I find the manuscript quite polished and ready for publication I do not think it is up to the level for Nature Communications.

Let me argue. The authors consider the core of the paper the fact that they have successfully managed to manipulate activity of rat somatosensory cortex and this caused alteration in time judgments of the rat. While this is a nice feat and shows that the authors have a nice platform for such kind of experiments, it should not surprise at all. The very same authors have already demonstrated elsewhere that if rats are stimulated with a higher intensity vibration they experience longer durations than usual. One of the things that happens for sure with such strong vibrations is that the firing rate in somatosensory cortex increases. What is happening here is that, the boost of firing rates is obtained via optogenetics and not via peripheral stimulation. But essentially is the same thing. It would be as making a big deal that stimulation of a neuron of V1 induces a phosphene. It's great that a lab has this technique but does not open new avenues for research.

What also strengthens my opinion (but I am willing to change my mind if the authors proved me that there is something substantial that I have missed) is that with the platform they have setup they could have addressed really novel research questions. For instance they could have followed the flow of information from somatosensory cortex onwards to see exactly which circuit downstream is involved in time perception. But it is not what is done here.

The only aspect of the paper which is a bit surprising is that manipulation during the first and second interval yields differential effects. And it matches behavioural data. But this effect in itself would require controls (e.g. what happened is rats made a 1 interval 2 alternative forced choice?) and it is not central to the manuscript, the way it is presented now.

This being said I only add two minor points. The first is that it wouldn't hurt to add some references to the literature that has demonstrated how intensity interferes with duration judgments (see for instance the saga in the visual modality that followed Kanai and Verstraten JOV 2008). The second is that the last sentence of the manuscript which aims at alluding that these findings can be relevant in schizophrenia research is a little bit of a stretch. In particular given that SZ has a complex etiology.

Reviewer #3 (Remarks to the Author):

Significance

Reinartz and colleagues provide a sophisticated yet elegant analysis of the role of a primary sensory area, S1 of a rat, in temporal perception. To do so, the authors combine behavioral analysis, optogenetic

manipulation, neural recording, and computational modeling. Overall, the effort is an important contribution to our understanding of the neural mechanisms that play a role in time perception.

Overview

The work presented by the authors have several compelling features. First, they perform detailed psychophysical analysis of rat behavior in a task asking them to discriminate between the durations of two tactile stimuli. Using an elegant task design, they are able to determine that the rodents learn to discriminate between interval durations and (for the most part) ignore the amplitude of the stimuli. Second, they perform optogenetic experiments to both increase and decrease the rate of neural responses in vS1. These causal manipulations appear to shift perception of duration to longer and shorter intervals (although see below). They then analyze the firing of neurons in vS1 during the task and make a convincing argument that the total spike count of neurons in vS1 is correlated with the perceived interval. Finally, they leverage the neural data they collected to model the population activity of vS1 and demonstrate that, when combined with a downstream integrator of the appropriate form, the properties of vS1 neurons can explain both their behavioral and optogenetic findings. For the most part the work is compelling, elegant, and should be an impactful addition to the field of timing perception. However, I do have some major concerns that should be addressed before publication.

Major Issues

1. First, I highly recommend that the authors simplify their presentation for the readers. All major analyses in the paper are based on the subset of stimuli used to generate the psychometric curves. Although not necessary, the paper would be improved if the authors focused the body of the paper on just the subset of trials used to generate the psychometric curves and simply state that animals were trained on the full stimulus set in the methods.
2. At points the conclusions of the paper are somewhat overstated. For example: "Time perception is thus as deeply intermeshed within the sensory processing pathway as is the sense of touch itself and the door is now open to unraveling the experience of time with the toolbox of sensory coding." Obviously, these results only apply to case where the intervals are filled with a stimulus and have little bearing on the perception of a duration demarcated by a brief stimulus at the beginning and end of an, otherwise empty, interval. While the authors concede as much in the discussion, the abstract and introduction should also be appropriately circumspect.
3. Related: it is worth noting that their "...hypothesis is that the ongoing activity within the primary sensory cortex plays a direct and systematic role in the judgment of time" has been suggested by a large body of experiments in human duration perception. For example, duration judgements dilate and contract according to the temporal frequency of a drifting grating presented during the interval (Kanai et al. 2006). These have led to models of sensory change that contribute to duration perception (Ahrens and Sahani 2011). This work deserved addressing in both the introduction and discussion.
4. Unfortunately, the results from the optogenetic manipulations, in particular those using eNphR3.0 to reduce firing rate, are not entirely convincing. The model presented by the authors makes a specific prediction in terms of the effect of optogenetic manipulation on the psychometric function - changes in the total number of spikes should change in the point of subjective equality (PSE). Based on the error bars presented in the figures, it appears that the effects are not significant (also, please see minor issues

below). Traditionally, one would improve the statistical power by using the fit of a logistic function to the psychometric data across data points to measure the shift in PSE. This appears to have not been sufficient, so the authors chose to perform an SVD analysis. It is not clear to the reviewer that this still correctly identifies only shifts in the psychometric function, and not more complicated effects that include shifting, changes in slope, differences in lapse rates, etc. I have two specific requests: first, simulate data where optogenetic manipulations generate different combinations of changes in PSE, slope, and lapse rate and then demonstrate that their method of assessing the significance of changes in behavior induced by optogenetic manipulations are only sensitive to the changes in PSE. Second, please include a summary of parameter fits of the model to allow the reader to assess how optogenetic manipulation of spike counts influenced the psychometric function.

Minor Issues

The methods for the stimuli and their generation are not entirely clear. At what rate were velocities sampled (e.g. once a ms)? Reporting the range of intensity standard deviations and durations is not sufficient when the values are not spaced regularly. Please be exact about the values used or the method used to select them.

Equations 3 and 4 - \exp -> \exp .

Paragraph starting at line 117 - it would be more straightforward to just to state that bias was measured as the the average change in perceived duration (intensity) as a function of the change in intensity (duration). Using 'likelihood' here suggests, at least to the reviewer, that a Bayesian analysis was performed.

A supplementary figure mimicking Figure 2A, left and Figure 2A, upper right, but to show the expression of eNpHR3.0 would be appropriate so the reader can assess the quality of transfection and the potential for off-site effects.

Figure 2C and 2D, middle: how, specifically, was the permutation test performed? This should be clearly explained in the methods.

Figure 2C and 2D, middle: was the standard error measured across rats?

The method for performing the SVD should be more specifically laid out in the methods. The reviewer assumes that only the estimates of the PSE and choice percentage data estimated from resampled data went into the classifier, but that should be clearly stated in the methods.

Figure 2E, middle: are the error bars plotted in this figure?

Figure 3B: as plot, this figure implies that the PSTH for each neuron was calculated for identical rod velocity traces (e.g. 'frozen noise' trials). If this was not the case, I suggest changing how the traces are presented, or at least clarifying in the legend

Have the authors considered the effect of correlated variability in the spike counts of neurons? That is, when one neuron tends to fire more spikes, other neurons also tend to fire more spikes (e.g. Zohary et. al. 1994). Do they think that this could explain the fact that variability of behavioral biases was much larger than the neurometric biases of their model?

Line 455: I don't think it is necessary to cite a COSYNE abstract when there are plenty of published studies on the neural mechanisms of "empty" interval timing (e.g. Mauk, Buonomano, Merchant, Jazayeri groups).

Line 484: It would be appropriate to cite the work on the effect of the order of interval presentations on time perception here (e.g. Dyjas, Bausenhart, and Ulrich).

The authors are grateful to the three reviewers for their careful reading of the study, their appreciation for its merits, and their insightful and constructive critiques. We apologize for the long period elapsed between receiving the feedback and the present resubmission. This is mainly due to the addition of new experiments, but the depth of the revisions was another contributing factor.

On the basis of the critiques offered by the three reviewers, the study has undergone major revisions. These include:

- new experiments that show photoexcitation-evoked expansion of perceived stimulus duration in a new psychophysical task, thereby generalizing the results,
- new and more rigorous analytical procedures and statistical testing,
- better layout of figures,
- clearer and more readable text, including discussion of the issues suggested by the reviewers.

In the point-by-point replies below, the reviewer comments are in blue font and our replies are in black font. In the resubmitted manuscript, text which has been revised and text which is completely new, with respect to the original submission, are in dark blue font. New statistical analyses are also in dark blue font. Several figures are reorganized (new panels, other panels moved to Supplementary, and so on).

REVIEWER COMMENTS

Reviewer #1 (Remarks to the Author),

Reviewer #1, point 1:

Combining optogenetic perturbation with a psychophysics discrimination task, Reinartz et al. investigate the contribution of vibrissal somatosensory cortex (vS1) activity to the perception of sensory stimulus duration and intensity. The authors train two groups of rats to either discriminate the duration or the intensity between pairs of vibrissal stimuli. They then use optogenetic photoactivation and photoinhibition of vS1 activity to assess changes in animals' perception of intensity or duration. Through extracellular electrophysiological recordings, the authors then examine neural codes for duration and intensity perception, informing a model to recreate observed psychophysical data. The authors aim to challenge a common belief that "the primary sensory cortex merely relays start and stop signals", and instead propose that "activity of vS1 plays a direct role in time judgement". However, because the optogenetic perturbation conducted in this study covered the entire duration of the stimulus, including the start and stop, the behavioural effects observed cannot rigorously differentiate the above hypotheses. To directly test them, optogenetic manipulation should be restricted to a central portion of the stimulus in which onset/offset triggers are unaffected. This study also demonstrates the potential to discriminate between sensory perception of intensity and duration through vS1 spike rate coding vs spike count coding respectively. However, the analyses shown in the study remain somewhat superficial. We propose further analysis to support this claim in detailed comments below.

Overall, this study provides evidence for the neural correlates of sensory duration perception in vS1 and that vS1 activity may be required for such perception. However, evidence in favour of the proposed models that vS1 activity provides a continuous input to a downstream time perception integrator, in contrast to only providing a stimulus onset and offset signals, are not strongly supported by the experimental data (as discussed above). Furthermore, we were concerned with the strong claims from the optogenetic perturbation experiments with relatively small effect sizes, and at times, missing control experiments. Finally, the wording of the manuscript can be improved for clarity as well as to more accurately reflect the topics investigated (stimulus duration rather than time perception). As a result, despite the potential interest in the underlying topic, we have limited enthusiasm for the publication of this study before these concerns can be addressed.

AUTHORS' REPLY, point 1:

We thank the Reviewer for their appreciation of the study's merits. With regard to the Reviewer's comment that "... we were concerned with the strong claims from the optogenetic perturbation experiments with relatively small effect sizes..." we note that the effect size of ChR2 in the present study is comparable to the effects reported elsewhere. For instance, one landmark, highly-cited paper (Hanks et al., 2006, Nature Neuroscience, 9: 682–689), used electrical microstimulation (a more invasive technique than optogenetics) in monkey LIP during a memory-guided saccade task to support the claim that "microstimulation of this cluster caused an increase in the proportion of choices toward the RF of the stimulated neurons. These results demonstrate that the discharge of LIP neurons is causally related to decision formation in the discrimination task." Their psychometric curve effect, which has since been taken as unquestioned proof of LIP's role in visuomotor decision making, is shown below (left and center plots) in comparison to the curve shift from the present manuscript (right plot).

Effect size can be quantified as the mean difference in choice for the stimulation vs the control condition. Using the Hanks et al. data, we compute an effect size of 2.9% for one monkey and 5.4% for the other. In our data, the same measure gives an effect size of 5.1% averaged across 5 rats.

If we use only the central point, corresponding to the condition in which an unbiased subject will split their choices evenly ($T_2 = T_1$ for our study and random motion direction for Hanks et al.), we compute an effect size of 2.2% for one monkey and 5.0% for the other. In our data, the same measure gives an effect size of 11.9% averaged across 5 subjects.

Similar exercises for other published work, including recent optogenetic studies in high-profile journals, confirm that the effect sizes in the present manuscript are not unduly small, especially considering that the aims were not to disrupt behavior but to subtly tune choices. In the revised manuscript, we have emphasized that the optogenetic perturbations left the trained behavior intact (lines 217-221):

If optogenetic manipulation had disrupted performance directly at the decisional level, it would be expected to affect choices no matter which stimulus pair was presented; instead, the manipulations merely shifted the percept in a systematic way, as borne out by the psychometric parameters (Supplementary Fig. S4).

We agree that the effect of optogenetic inhibition on rats' duration judgment is comparatively small and have revised the manuscript's organization and interpretations accordingly. In more detail, because

- Halorhodopsin (eNpHR3.0) is a slow active-pump opsin, and is known to require stronger light power as compared to that required to achieve ChR2 excitation effects (ChR2: Boyden et al., 2005; eNpHR3.0: Gradinaru et al., 2010), and
- the maximum light power of the orange (eNpHR3.0) LED was lower than that of the blue (ChR2) LED,

we now consider it more appropriate to present the Halorhodopsin condition as a control rather than (as in the original submission) an active intervention for direct comparison to ChR2. Halorhodopsin is a less potent intervention, unlikely to generate effects of equivalent size.

However, the orange LED optic input could have produced tissue heating and increased neuronal firing; in that case, the perceived stimulus duration might have increased. Instead, the Halorhodopsin condition led to a systematic trend for a *reduced* stimulus duration percept, albeit statistically weak. The value of the Halorhodopsin experiment must be seen, therefore, as a contrast to and control for Chr2 rather than as a principal positive finding. As such, we have moved some parts of those results to Supplementary sections and have presented the ChR2 condition as the main result supporting the hypothesis of vS1 involvement in stimulus duration perception. Details on this reorganization are provided later in the context of other comments. We are grateful for the Reviewer's attention to the Halorhodopsin results, which led us to put them in the proper framework.

We would like to address the terminology of "stimulus duration" versus "time perception." In the literature, many behavioral measures of time – the unfilled time between start/stop events, the motor reproduction of a previously perceived duration, the judgment of two events as being synchronous/asynchronous – are all referred to as time perception, in addition to the specific paradigm through which time is perceived. Because the present paradigm, judgment of stimulus duration, is one of the several types of episodes that comprise time perception, we think the original terminology is acceptable. Following the Reviewer's suggestion, we did change the title to make the specific time perception paradigm more visible: it is now ***Direct contribution of the sensory cortex to the judgment of stimulus duration*** (previous title: ***Direct contribution of the sensory cortex to time perception***). This is addressed again under point 12.

The other issues raised by the Reviewer under point 1 are repeated in subsequent points and we address them as they arise.

Reviewer #1, point 2:

Major concerns:

1) Design, implementation and interpretation of the optogenetic perturbation experiments

Design: The main conclusion of the paper rests on the optogenetic perturbation experiments. In particular, the authors aim to challenge the hypothesis that "the primary sensory cortex merely relays start and stop signals", and instead propose that "activity of vS1 plays a direct role in time judgement". Although the results shown in the paper are consistent with the latter hypothesis, it is also not inconsistent with the former hypothesis. To directly tease them apart, a key experiment should be optogenetic perturbation where the laser is restricted to a central portion of the stimulus in which onset/offset triggers are unaffected. We believe that this experiment is needed to support the main conclusion of the manuscript.

AUTHORS' REPLY, point 2:

The reviewer is likely referring to the phrase "the ongoing activity of the primary sensory cortex plays a direct and systematic role in the judgment of time." We understand the reviewer's concern; although we do not have data where the laser is restricted to a central portion of the stimulus, we have instead added two other pieces of evidence that, in our view, speak more directly to the question of whether vS1 plays a role in time judgment by merely relaying start and stop signals. First, (lines 324-333):

One plausible mechanism by which vS1 might contribute to the perception of elapsed stimulus time is by providing downstream centers with onset and offset times that could function similarly to a stopwatch. The increase in perceived duration with photoexcitation, in this scenario, would result from a greater elapsed time interval between onset and offset. This model is tested in Fig. 3h. Onset and offset of the vibration-evoked responses were detected when the firing rate crossed a threshold, set as 1 STD of change from the mean firing rate. In the left panel, neuronal population onset and

offset times for vibrations of 694 ms are shown as vertical lines under the light-on condition (blue) and light-off (black) conditions. Nearly identical temporal boundaries are recovered under the two conditions, symbolized by the blue and gray stopwatches which record the same elapsed time. The above analysis continues to population quantification (Fig. 3h, right panel).

We originally pointed to the accumulated, weighted spike count as a potential mechanism. This is confirmed by using the same firing rate onset/offset times (lines 343-353):

The alternative model, where perceived duration is related to the integration of vS1 firing, is given in Fig. 3i. The plots of the left panel illustrate the firing feature that could underlie this coding mechanism, the total (filled) area under the PSTH. By boosting vS1 excitability, the total number of spikes transmitted to downstream integrators, for any given actual vibration duration, is increased under photoexcitation. This accumulation of vS1 activity is symbolized by the hourglasses, where the “readout” records a greater duration with light-on (lower plot) versus light-off (upper plot). In the right panel, the accumulated spikes between response onset and offset are plotted under the light-off (abscissa) and light-on (ordinate) conditions. Similarly to the plot in Fig. 3f, greater accumulated spike counts between estimated onset and offset are seen for any given stimulus duration under the light-on condition. As in the left panel, this accumulation of vS1 activity is symbolized by the two hourglasses.

As a second test to determine whether the results depend on the temporal profile of optogenetic stimulation, we have conducted new experiments in which vS1 was optogenetically excited by a *stochastically varying light* instead of a strong, continuous light signal (as in the original data set). This presumably adds variability to vS1 response onset and offset times and makes the on-off signals no longer predictive of total stimulus duration, since light intensity fluctuates continuously. These new results (n = 4 rats) show comparable overestimation of perceived tactile vibration duration (new Supplementary Figure S5). (Authorship now includes Maria Ravera, who contributed to the new experiments.) Lines 198-204:

EYFP-ChR2(H134R)-expressing rats (n = 4) trained to compare the duration of each trial’s single vibration duration to a fixed, reference duration showed a bias towards longer perceived duration on trials with photoexcitation (Supplementary Fig. S5). Here the light was modulated stochastically across the tactile stimulus presentation. This finding indicates that the vS1 role in stimulus duration perception is not specific to the delayed comparison working memory task, nor is it specific to the profile of light application in optogenetic trials.

In sum, that changes in perceived stimulus duration might arise through start/stop signals transmitted through vS1 is a valid hypothesis but it is not, upon testing, supported by re-analysis of the original data nor by the new data.

Reviewer #1, point 3:

Control: To control for the non-neural-activity related effects of laser stimulation (including tissue heating, retinal activation during photo-stimulation etc), eYFP-control experiments should be carried out for photostimulation and photoinhibition experiments.

AUTHORS' REPLY, point 3:

We understand the reviewer's concerns and the motivation to control for tissue heating and retinal activation. However, following the 3R principles (Replacement, Reduction and Refinement) that must be adopted for animal wellbeing, which compel us to limit the number of rats used, we employed the existing data to achieve two controls, while for a third control we did new experiments (also mentioned under point 2, above):

(1) In experiments reported in the original manuscript, to control for **retinal activation** as a potential factor underlying perceptual biases, we provided identical blue light signals visible externally while the rats performed duration judgments (Supp. Fig S2, now main Figure 2d). If rats used retinal activation originating in transmission through the brain to judge vibration duration on optogenetic trials, then the external retinal activation by direct light likely would have caused even more pronounced behavioral effects. Instead, the blue light psychometric curve overlaid the no-light curve.

(2) While we used optogenetic vS1 excitation (ChR2) during tactile stimulus delivery to modulate neuronal coding (analyzing their effects on neuronal spiking and modeling the effects with respect to time perception), we also used vS1 inhibition (eNpHR3.0) in what we now present as a control condition (previously Figure 2b, now Supplementary Figure S2a). If blue light had **heated up the tissue** to create the expanded stimulus duration effect in the vS1 excitation experiments, one would also expect to find duration over-estimation with red light. Even if the red light sessions were to cause an effect of a different magnitude due to lower wavelength energy, surely the sign of the change – longer perceived duration – would be the same as that caused by blue light. However, the control condition gave a weak but significant rightward shift indicative of a compression of perceived duration. Red light appeared to yield an effect consistent with opsin-mediated neuronal inhibition, but inconsistent with the excitation that would be caused by heating.

(3) We added a new experiment, providing behavioral data of rats ($n = 4$) performing categorization of tactile vibration duration. The new rats were trained in reference memory (RM), not working memory (WM). This offers the advantage of testing the generalization of the main hypothesis to a new task. These RM experiments were combined with optogenetic vS1 excitation by blue light that stochastically varied in amplitude across the stimulus duration. This stochastic light amplitude variation, in comparison to the continuous, uniform light applied in the original experiments, delivers considerably weaker total optic power and thereby would be expected to create less tissue heating. Still, a significant bias towards duration

overestimation in vS1 excitation trials was observed (see Supplementary Figure S5, inserted below into this reply letter).

Finally, it is worth considering how potential tissue heating (even if preceding arguments make it seem unlikely) would affect the manuscript’s main conclusion. If vS1 firing were in fact modulated by increased temperature rather than by the intended optogenetic mechanisms, the principal claim – that vS1 neuronal firing is directly read-out and integrated as one mechanism underlying time perception – would still stand.

Reviewer #1, point 4:

Presentation and interpretation: In Fig. 2B-D, the authors mainly compared the psychometric curves across different perturbation conditions. What about the effect of each experiment relative to the control condition? “No light” control psychometric curves should also be plotted for each of these panels. This would be especially illustrative for manipulation during stimulus 1, where the effect size is small. For clarity, significance should be determined between these control curves and the experimental curves, alongside significant difference between centre points. It would also be helpful if the authors can quantify the multiple parameters related to the change in the overall psychometric curves (lapse, slope, intercept etc) rather than just focusing on centre points.

AUTHORS’ REPLY, point 4:

We did three major revisions in reply to this point:

- change in the treatment of the inhibitory optogenetic experiments,
- change in the treatment of stimulus 1 effects,
- more complete presentation of psychometric parameters.

In detail: In recognition of the importance of the *No light* control condition, we moved Figure 2e – the summary of optogenetic excitation, inhibition and their control (“no light”) conditions – up to panel 2b (and Supplementary Figure S6a) and improved the statistical testing (see also new Supplementary Figure S4). As an improvement in significance testing (2b-e, middle) we now applied the permutation test (1000 resamples), not merely on the center point, but on the overall percentage of choice, including all data points apart from the two “easiest” stimulus pairs, where strong sensory evidence would be expected to override any possible optogenetic effect. Regarding interpretation of the size of the effects, we must emphasize again that this was not a gain- or loss-of-function study – the experiments were designed to search for graded

effects of primary sensory cortex firing modulation within the context of the rats' continued performance of a difficult and complex behavioral task (also see reply to point 1). We note that only a handful of studies have shown successful training of rats in tactile working memory and only our research group has experience, to date, in tactile duration working memory (Toso et al., 2021a, 2021b). We find that distractions always interrupt the rat's performance of the task, an effect we attribute to the high cognitive load arising from the memory components of the behavior. Thus, if the rats had sensed a tactile anomaly originating in vS1 optogenetics, they might have qualitatively changed their strategy, e.g. reverting to random reward spout selection. Optogenetic manipulation can in itself bring many nonspecific side effects (Andrei et al., 2021; Owen et al., 2019; Wolff and Ölveczky, 2018); motivated by our attempt to keep the rats unaware of light trials being different from no-light trials, we applied only mild optogenetic excitation. And just as intended, optogenetic excitation did not disrupt performance, but merely shifted the percept in a systematic way.

As stated under point 1, a major change in the revision is to focus (both in analysis and modeling) on the optogenetic excitation. The revised manuscript reduces the statements related to the optogenetic inhibition experiments.

Comparison between light-on stimulus 1 to the no-light condition showed smaller effects, which we assign to the well-known phenomenon of contraction bias (Akrami et al., 2018; Esmaeili and Diamond, 2019; Fassihi et al., 2014; Raviv et al., 2012). Contraction bias in delayed comparison tasks causes the memory of the perceived quantity of stimulus 1 (here, duration) to shift during the interstimulus interval towards the midpoint of the recent history of stimuli. One of the main outcomes of contraction bias is a reduction in the weight of stimulus 1 in the final choice, inasmuch as variation of stimulus 1 affects the final decision less than does variation of stimulus 2.

As requested, we further quantified the psychometric curve fitting parameters (new Supplementary Figure S4, inserted below into this reply letter) and found a significant difference in the bias parameter (μ) between vS1 excitation during stimulus 1 versus stimulus 2. This measure confirms the horizontal shift in the psychometric curve, suggesting a significant change in perceived duration. The other fitting parameters, slope (σ) and the two lapse parameters (γ , λ) showed no significant difference between excitation during stimulus 1 versus stimulus 2. In other articles reporting psychophysical effects of optogenetic intervention (e.g. Znamenskiy and Zador, 2013), perceptual biases are often paralleled by changes in lapse and/or slope parameters, indicating that intended outcome may in fact be mixed with factors related to attention, motivation, or distraction. In the present study, the ensemble of positive and negative effects bears out our intention to test for modulation of the stimulus duration percept without degrading performance: even though stimulus 2 vS1 excitation affected choices more than did stimulus 1 vS1 excitation, that effect did not play out in slope or in lapses. The fitting parameters for vS1 inhibition during stimulus 1 versus stimulus 2 did not uncover significance in any of the parameters, though the μ parameter (perceptual magnitude shift) shows a trend opposite to vS1 excitation. The opposite sign indicates that the tissue was not excited by heating and is more consistent with opsin-mediated inhibition. As discussed before, due to experimental limitations, given the properties of Halorhodopsin and the reduced light, it was inappropriate to originally interpret the inhibition condition as equivalent in effect to the CHR2 condition. We could identify weaker perceptual effects for vS1 inhibition, and we now interpret the inhibition results as a control. Nevertheless, when the analysis is restricted to rats' choices with identical stimulus 1 and 2 duration ($\Delta T = 0$, both stimuli of 334 ms duration), a significant choice bias even with optogenetic vS1 inhibition emerges (old Figure 2c, middle panel; new Supplementary Figure S2a).

Reviewer #1, point 5:

Finally, we were quite surprised to see that the authors used constant light delivery at relatively high laser power levels for the ChR2 experiments, which may lead to tissue heating and unphysiological activation of cells. Light power should be reported as a range as opposed to '>=10 mW', and justification given for the chosen light power and light stimulation profile. Providing pulsed photostimulation at frequencies within the physiological range for vS1 neurons would reduce potential tissue heating and opsin deactivation.

AUTHORS' REPLY, point 5:

As an answer to the Reviewer's request, we used stochastic light for optogenetic vS1 excitation in a new set of experiments (see Supplementary Figure S5), showing that the results can also be obtained with non-constant light. Also please see the reply to point 3, above.

We added information about the range of applied light power, 10-12 mW, to the Methods text. In general, off-target effects resulting from high light power depend on the light density (mW/mm²). The experiments

reported here use a relatively large optic fiber diameter (\varnothing : 230 μm), resulting in a lower light density per light output power, as compared to thinner diameters used in other studies. The long intervals between the instances of light delivery also suggests that heat would not be accumulated during the course of a session. Because only about 100 trials in a session of 300-400 trials included light stimulation, and light was turned on for only a single stimulus per trial, the typical interval between light applications was 30-40 seconds.

A justification to use continuous versus pulsed photostimulation, is that the current study depends not only on perceptual biases, but also on the neuronal coding. With pulsed photostimulation we expected photovoltaic artifacts (Cardin et al., 2010; Mikulovic et al., 2016) to possibly deteriorate our read-out of neuronal firing.

Reviewer #1, point 6:

2) More detailed description and quantification of vS1 neural population in light on/off conditions:

As shown in Figure 3A, there are two groups of cells identified in vS1, light- and vibrissal -responsive neurons. Little exploration of the ‘heterogeneous’ recorded vS1 population in EYFP-ChR2(H134R) rats is presented besides 2 example neurons in Fig. 3A. Further analysis of other neuron clusters and their response to photostimulation would be helpful to identify how perturbation affects vS1 population activity. The authors should at least show more individual neuron examples in Supp figures, and provide population summaries beyond just Fig. 3B/C.

AUTHORS’ REPLY, point 6:

We point the reviewer to the additional characterization of neuronal responses and the modulation of firing rate, depending on tactile and optogenetic stimuli given in the Supplementary Figures S7 and S8. We added PSTHs of further example neurons and clustering based on the model in new Supplementary Figure S13 (see below).

Reviewer #1, point 7:

The firing rate profile in Fig. 3C of vS1 population activity during vibrissal stimulation shows an initial sharp increase in firing rate followed by a lower sustained firing until stimulus offset. The vibrissal stimulation plus photostimulation condition shows an almost identical pattern with a higher firing rate in the initial peak. This is reported as an evenly distributed boost in the vibrissae-evoked response (L266), but it appears that the increase is largely limited to the first 100 ms. The authors should provide an explanation as to why this initial peak might be the prevailing feature in light-on and light-off conditions and why the primary difference between light-on and light-off conditions is a change in magnitude of the initial peak.

AUTHORS' REPLY, point 7:

The initial peak in tactile responses recorded in vS1 is a feature that has been reported in publications beginning in the 1980s and can be explained by an initial unimpeded “window of opportunity” (reviewed in Bruno, 2011) for excitatory thalamocortical input which is quickly dampened by feed-back inhibition and neuronal adaptation. We added three sentences to clarify the speculated dynamics in the Discussion (lines 483-489).

In our hands, photoexcitation-evoked changes in the vS1 response to vibrissal vibration were seen in the peak of the initial response and, to a lesser extent, in the steady-state response (Fig. 3c). This suggests that photoexcitation acts most prominently at response onset, when stimulus-evoked responses are believed to propagate through local cortical circuits before strong feedforward inhibition clamps down on the excitability of sensory-coding neurons⁴¹. The early response of sensory cortex appears to comprise part of the stream that is accumulated at successive stages of processing.

The data suggest that optogenetic-evoked changes in population firing are not entirely confined to the initial peak. Figure 3a, left panel, shows one unit that is excited by light across the entire stimulus period. Careful inspection of Figure 3b also indicates that neurons can show sustained optogenetic excitation effects, as described in new text (lines 276-279):

After an early peak, firing rate remained stable until offset. The right panel shows the PSTHs of the same population under photoexcitation, revealing a modest boost in the vibrissae-evoked response, particularly at vibration onset and thereafter distributed evenly across the period of tactile stimulation.

We agree that Figure 3c is not optimized for focusing on the characteristics of the optogenetic effect. In the new Figure panel 3h (left) and 3i (left), we show the change in firing rate to the longest duration (694 ms) along the entire stimulus presentation.

Reviewer #1, point 8:

Similarly, recording firing rate in fig. 3E during the final 100 ms of stimulus presentation removes information about this initial peak from analysis, despite it being included in the count coding analysis of fig. 3F (we computed counts from vibrissal-stimulus onset to offset (L291)). This choice should be consistent across analysis and maximize information contained within the population activity.

AUTHORS' REPLY, point 8:

If firing rate (or accumulated spike count, for that matter) is intrinsic to the duration code, the onset peak (identical for all stimulus durations), cannot be the key response feature inasmuch as the stimulus does not have a final elapsed duration at the onset. The coding feature must be *explicit* at the end of the stimulus. Figure 3e is intended to test whether there are features in the firing rates at the end of the stimulus coding correlated with stimulus duration; the analysis shows that firing rate would not constitute a robust coding mechanism (by contrast, it would offer a robust code for vibration intensity; Figure 3d). For completeness,

as asked by the Reviewer, we added a figure comparing firing rates along the entire stimulus durations to Supplementary Figure S11.

Reviewer #1, point 9:

Additionally, there is a sharp decrease in spike rate at the point of stimulus offset in both light-on and light-off conditions that appears to be a break in the traces. Can the authors explain why we are seeing such a sharp, almost instantaneous decline in activity rather than a gradual decrease of spike rate after light offset?

AUTHORS' REPLY, point 9:

This is explained by the shape of the filter that converts spikes as points processes to firing rate (explained in Methods, lines 778-784). With the new Figure panels 3h (left) and 3i (left), the PSTHs are also represented without the sharp activity decline.

Reviewer #1, point 10:

3) Trial-by-trial neural-behavioural correlation analysis to support the count coding hypothesis
In Fig. 3D-G the authors show evidence that support a rate coding of intensity and a count coding of duration. It would be great if the authors can conduct some trial-by-trial neural-behavioural correlation analysis to further support these mechanisms. Specifically, across trials with the same stimulus duration, does the trial-by-trial fluctuation in spike count correlate with the probability of behavioural report? Similarly, across the trials with the same stimulus duration, does the trial-by-trial fluctuation in spike rate not correlate with choice probability? These analyses can provide further evidence for the count coding of duration in vS1.

AUTHORS' REPLY, point 10

This would be an informative analysis. Unfortunately the data set does not include a sufficient number of neurons per recording session to permit a trial-by-trial correlation analysis. Our concern is that the measured trial-by-trial fluctuations might not sufficiently represent the trial-by-trial fluctuations integrated by the read-out area and would lead to “overfitting” in the interpretation of the analyses.

Reviewer #1, point 11:

4) Wording/writing can be improved for clarity and accuracy:

AUTHORS' REPLY, point 11:

The authors have worked to improve clarity and accuracy. The improvements in writing that did not relate to changes in content are left in the original black font.

Reviewer #1, point 12:

The suggested title implies that this study gives evidence for a role of the somatosensory cortex in general time perception, which may be misleading. The manuscript focused on studying the perception for the duration of a tactile stimulus, which requires tracking of salient sensory cues. This process may be different from a sensory-modality-independent time perception. Consequently, we suggest that the authors update their title/abstract accordingly and maintain this clear distinction between time perception and sensory input duration throughout the paper.

AUTHORS' REPLY, point 12:

Although we maintain that stimulus duration perception is one of several forms of time perception (see point 1), we do agree that a more specific title is helpful, and the revised title is:

Direct contribution of the sensory cortex to the judgment of stimulus duration

Reviewer #1, point 13:

In general, the manuscript can be improved in terms of the clarity of the writing. For example, many parts of the Results section reads like part of the Figure legend. Rather than describing what is plotted in a figure, it would be helpful for the readers if the authors could instead describe and synthesize the results.

Minor concerns:

1) Only averaged psychometric curves are shown throughout the paper, with only a few showing error bars. Individual psychometric curves should be shown to reveal variance across individual rats. In addition, when the authors quantify the effects of optogenetic perturbations across animals, individual differences should be taken into consideration as well as the average main effects (e.g. using mixed effects in GLME).

AUTHORS' REPLY, point 13

Perceptual biases of individual rats are shown in old Figure 2E, right panel (now Supplementary Figure S6a). We added Supplementary Figure S3, showing the psychometric curves of the individual rats. We further added Supplementary Figure S4, showing the fitting parameters depending on optogenetic vS1 excitation and inhibition.

Reviewer #1, point 14:

2) Fig. 2: What happened after optogenetic inhibition for intensity rats? Have the authors conducted this experiment for comparison?

AUTHORS' REPLY, point 14:

As this study is focused on time perception, training an additional set of intensity rats to undergo optogenetic inhibition was not in the experimental design. Since optogenetic inhibition effects in duration rats were weaker than excitation effects and therefore not directly comparable, we now interpret inhibition as a control condition, as explained above. Such a control in intensity rats would be beyond the scope of this study.

Reviewer #1, point 15:

3) Fig. 2: Only two subjects are used for inhibition in duration rats. Considering that the results are somewhat noisy, it might be important to increase this sample size.

AUTHORS' REPLY, point 15:

As discussed earlier we see the optogenetic vS1 inhibition more as a control to vS1 excitation and redimensioned our statements regarding inhibition. Unfortunately, we were not able to train/test new rats in working memory.

Reviewer #1, point 16:

4) Fig. 2E: missing bar for control rats (Fig. 2E right)? It does not seem like that the data is averaging at exactly 0?

AUTHORS' REPLY, point 16:

The bar for the control rats is indeed very close to zero and therefore not visible. We have changed the visualization in Figure 2e (right), now Supplementary Figure S6a to better illustrate the average for control rats as well.

Reviewer #1, point 17:

5) The high variability in control rats shown in figure 2E (right) makes it unclear whether bias in T perception is actually significantly different between perturbation vs control conditions? This seems inconsistent with the average curve results in 2E middle. Is this because most of the data (trials) come from a few animals?

AUTHORS' REPLY, point 17:

While the psychometric curves in Figure 2e (middle), now 2b (middle), are generated by control ("no light") trials of rats during optogenetic perturbation sessions (Chr2, eNpHR3.0), Old Supplementary Figure 2a (right, new Figure 2d) shows experimental results from a different condition. The same rats performed the duration discrimination task, but instead of delivering the light through the optic fiber to vS1, a strong light with identical wavelength was presented externally as a visual input (psychometric curves now shown in Figure 2d). We improved the wording in the Results (lines 164-173).

Reviewer #1, point 18:

6) Supp. Fig 2A (middle): External light control rats show a much stronger bias to choosing $T2 > T1$ than would be expected from Fig. 1C. Does this bias remain in control rats when the LED is not present? The psychometric curve of the no light condition for these rats should be plotted alongside the external light curves; and any difference between the external light and no light conditions should be acknowledged.

AUTHORS' REPLY, point 18:

We cannot fully grasp the intention of this comment from the reviewer. Figure 1c describes the bias that tactile vibration exerts on duration perception taken from a different set of rats. Supplementary figure 2a (middle), the experiment with the external light control (now Figure 2d), was done on the same set of rats used for optogenetic vS1 excitation (Figure 2b, now 2c) and inhibition (Figure 2c, now Supplementary Figure S2a). As visible in all these figures, taken together the rats have a slight tendency to judge $T1 > T2$. Such biases fluctuate from rat to rat and are beyond the control of the experimenter; we believe that these small biases do not affect results and interpretations.

Reviewer #2 (Remarks to the Author):

Dear All,

I have read with great care the submission by Reinartz et al, and while I find the manuscript quite polished and ready for publication I do not think it is up to the level for Nature Communications.

Reviewer #2, point 1:

Let me argue. The authors consider the core of the paper the fact that they have successfully managed to manipulate activity of rat somatosensory cortex and this caused alteration in time judgments of the rat. While this is a nice feat and shows that the authors have a nice platform for such kind of experiments, it should not surprise at all. The very same authors have already demonstrated elsewhere that if rats are stimulated with

a higher intensity vibration they experience longer durations than usual. One of the things that happens for sure with such strong vibrations is that the firing rate in somatosensory cortex increases. What is happening here is that, the boost of firing rates is obtained via optogenetics and not via peripheral stimulation. But essentially is the same thing. It would be as making a big deal that stimulation of a neuron of V1 induces a phosphene. It's great that a lab has this technique but does not open new avenues for research.

What also strengthens my opinion (but I am willing to change my mind if the authors proved me that there is something substantial that I have missed) is that with the platform they have setup they could have addressed really novel research questions. For instance they could have followed the flow of information from somatosensory cortex onwards to see exactly which circuit downstream is involved in time perception. But it is not what is done here.

AUTHORS' REPLY, point 1:

We thank the reviewer for their appreciation of the study's merits. First we address the lack of "surprise" generated by this study. Our earlier work on the stimulus intensity bias in duration perception makes a prediction and the present paper follows it up with more targeted methods; neuroscientific advances are meant to evolve in this way. Even if one were to accept from preceding work, without need for further proof, that vS1 is the input stage for the perception of stimulus duration, the question remains: Which are the firing features of vS1 that become, with subsequent processing, the percept of elapsed time? As a reminder, pasted in below is a relevant section in the Introduction (line 78-85) that addresses the need to specify not only the fact that sensory cortex is part of the time perception system, but also the underlying coding mechanisms:

In sum, here we seek to determine whether the tactile sensory cortex is part of the neuronal substrate for perceived time and, if so, what are the features of firing that causally lead to shifts in perceived time. The coding algorithms for sensory features are well established¹⁹⁻²⁵. Guided by these algorithms, we tested whether the effects of optogenetic manipulation on duration judgment could be predicted using real neuronal spiking patterns as input. The successful implementation of a computational framework for the perceived duration of tactile stimuli based on the firing patterns evoked by those stimuli opens up the field of time perception to the tools of sensory coding.

The current study directly manipulates vS1 to elucidate how the neuronal representation therein underlies the percept of elapsed time and rules out possible alternatives of the behavioral effect of stimulus intensity on perceived duration: for instance alertness and attention (Brown, 1997; Penney et al., 2000; Zakay and Block, 1996), integration *not* of sensory cortex but of alternative somatosensory pathways like the superior colliculus or thalamo-striatal pathways (Matell et al., 2003; McHaffie et al., 2005; Monteiro et al., 2023; Murray and Escola, 2017) or alternative vS1 sensory processing functions such as the relay of stimulus start and stop signals (see Reviewer #1).

In the revised paper, we make this motivation clearer and more explicit. For instance, we now include a new analysis to test existing theories about primary sensory areas being merely a relay station for starting and termination of a temporal event (Reviewer #1, and Li et al., 2021; discussed in Balasubramaniam et al., 2021; Meck and Benson, 2002). If vS1 were in fact merely relaying start and stop signals, then the increase of perceived duration in ChR2 optogenetic trials would be explained by a lengthening of the elapsed time between neuronal firing onset and offset (firing rate crossing the threshold of 1 STD from the mean firing rate) as compared to the same measure on non-opto trials. In contrast to that prediction, the analyses failed to find an extended response onset-to-offset duration with optogenetic excitation of the firing rate (new Figure 3h).

We originally pointed to the accumulated, weighted spike count as a potential mechanism. When the onset-to-offset durations extracted above were taken as temporal windows for spike counts, the analysis now reveals a small but significant spike count increase in trials with optogenetic excitation (new Figure 3i). The new findings of Figure 3, together with those carried over from the original submission, argue for the

accumulated spike count as a potential mechanism for duration perception, and firing rate as a potential mechanism for intensity perception.

Responding to the Reviewer's constructive spirit, we accept the invitation to argue: we object most vehemently to their portrayal of the present work as equivalent to finding "that stimulation of a neuron of V1 induces a phosphene." Phosphenes from visual cortical stimulation are widely known to constitute the most basic sensory phenomenon, a phenomenon that resides within the very same sensory modality to which the cortical region is bound. The phosphene analogy is, very roughly, related to our Intensity perception control. Yet even that analogy is weak, inasmuch as intensity is a finely tuned, psychophysically accessible measure, while phosphenes are unquantifiable visual impressions. Our main result is quite different: the involvement of tactile sensory cortex in TIME, a percept that is above and beyond the known sensory function of vS1. Furthermore, tactile sensory cortical involvement in time is not revealed as an "impression" (like a phosphene) but as a psychometrically documentable percept.

Reviewer #2, point 2:

The only aspect of the paper which is a bit surprising is that manipulation during the first and second interval yields differential effects. And it matches behavioural data. But this effect in itself would require controls (e.g. what happened is rats made a 1 interval 2 alternative forced choice?) and it is not central to the manuscript, the way it is presented now.

AUTHORS' REPLY, point 2:

The less pronounced behavioral effect of optogenetic excitation vS1 during stimulus 1 as compared to stimulus 2 can be assigned to the well-known phenomenon of contraction bias (Akrami et al., 2018; Esmaeili and Diamond, 2019; Fassihi et al., 2014; Raviv et al., 2012). As discussed in the context of Reviewer #1, point 4, contraction bias in delayed comparison tasks causes the memory of the perceived quantity of stimulus 1 (here, duration) to shift during the interstimulus interval towards the midpoint of the recent history of stimuli. One of the main outcomes of contraction bias is a reduction in the weight of stimulus 1 in the final choice, inasmuch as variation of stimulus 1 affects the final decision less than does variation of stimulus 2.

However, as mentioned by the reviewer, the relative behavioral weighting of the two stimuli is not central to the current manuscript. Indeed, as noted under Reviewer #1, point 2, new data were generated by training rats to compare a single vibration duration to a reference duration (reference memory (RM) task) rather than the working memory (WM) task of the original submission. Demonstrating generalization of vS1 effect on perceived stimulus duration in a different paradigm, with a different optic pattern, strengthens the main arguments of the manuscript.

Reviewer #2, minor points 3 & 4:

This being said I only add two minor points. The first is that it wouldn't hurt to add some references to the literature that has demonstrated how intensity interferes with duration judgments (see for instance the saga in the visual modality that followed Kanai and Verstraten JOV 2008). The second is that the last sentence of the manuscript which aims at alluding that these findings can be relevant in schizophrenia research is a little bit of a stretch. In particular given that SZ has a complex etiology.

AUTHORS' REPLY, minor point 3:

We added further citations on the interaction between stimulus features (e.g., intensity) and perceived duration, for several sensory modalities, including the visual system (Berglund et al., 1969; Ekman et al., 1969; Kanai et al., 2006; Stevens and Hall, 1966) to the Introduction part (line 65).

AUTHORS' REPLY, minor point 4:

We argue that the last sentence of our manuscript: “The generality of the model raises the prospect that anomalous sensory coding mechanisms may be one contributing factor in the time misperception at the core of multiple psychiatric disorders” is not too much of a stretch. Biases and distortions in time perception are well documented under specific neuropsychiatric conditions (Allman et al., 2011; Brock et al., 2002; Honma et al., 2019; Stevenson et al., 2017; Ueda et al., 2018) and it is not impossible that there could be non-normative processes governing the integration of sensory cortical firing by other brain regions.

Reviewer #3 (Remarks to the Author):

Significance

Reinartz and colleagues provide a sophisticated yet elegant analysis of the role of a primary sensory area, S1 of a rat, in temporal perception. To do so, the authors combine behavioral analysis, optogenetic manipulation, neural recording, and computational modeling. Overall, the effort is an important contribution to our understanding of the neural mechanisms that play a role in time perception.

Overview

The work presented by the authors have several compelling features. First, they perform detailed psychophysical analysis of rat behavior in a task asking them to discriminate between the durations of two tactile stimuli. Using an elegant task design, they are able to determine that the rodents learn to discriminate between interval durations and (for the most part) ignore the amplitude of the stimuli. Second, they perform optogenetic experiments to both increase and decrease the rate of neural responses in vS1. These causal manipulations appear to shift perception of duration to longer and shorter intervals (although see below). They then analyze the firing of neurons in vS1 during the task and make a convincing argument that the total spike count of neurons in vS1 is correlated with the perceived interval. Finally, they leverage the neural data they collected to model the population activity of vS1 and demonstrate that, when combined with a downstream integrator of the appropriate form, the properties of vS1 neurons can explain both their behavioral and optogenetic findings. For the most part the work is compelling, elegant, and should be an impactful addition to the field of timing perception. However, I do have some major concerns that should be addressed before publication.

Reviewer #3, point 1:

Major Issues

1. First, I highly recommend that the authors simplify their presentation for the readers. All major analyses in the paper are based on the subset of stimuli used to generate the psychometric curves. Although not necessary, the paper would be improved if the authors focused the body of the paper on just the subset of trials used to generate the psychometric curves and simply state that animals were trained on the full stimulus set in the methods.

AUTHORS' REPLY, point 1:

We thank the reviewer for their appreciation of the study's merits. We have simplified the wording and explanations, however, we chose to keep the information illustrated in Supplementary Figure 1, as our working memory experience informs us of the importance of using stimulus pairs that cannot be solved by reference memory (variable stimulus 1 and 2 durations) to make sure rats perform the delayed comparison. All the other illustrations are already limited to the subset of stimuli to generate the psychometric curves.

Reviewer #3, point 2:

2. At points the conclusions of the paper are somewhat overstated. For example: “Time perception is thus as deeply intermeshed within the sensory processing pathway as is the sense of touch itself and the door is now open to unraveling the experience of time with the toolbox of sensory coding.” Obviously, these results only apply to case where the intervals are filled with a stimulus and have little bearing on the perception of a duration demarcated by a brief stimulus at the beginning and end of an, otherwise empty, interval. While the authors concede as much in the discussion, the abstract and introduction should also be appropriately circumspect.

AUTHORS’ REPLY, point 2:

With the new title: **Direct contribution of the sensory cortex to the judgment of stimulus duration**, following Reviewer #1 request, the perception of “empty” intervals is already excluded. The Abstract (line 37-38) was changed from
“Perceived duration was dilated and compressed by optogenetic excitation and inhibition, respectively.”
to
“Perceived duration was dilated by optogenetic excitation.”

Throughout, we have revised and taken care to make the conclusions more specific to tactile stimuli whenever relevant.

Reviewer #3, point 3:

3. Related: it is worth noting that their “...hypothesis is that the ongoing activity within the primary sensory cortex plays a direct and systematic role in the judgment of time” has been suggested by a large body of experiments in human duration perception. For example, duration judgements dilate and contract according to the temporal frequency of a drifting grating presented during the interval (Kanai et al. 2006). These have led to models of sensory change that contribute to duration perception (Ahrens and Sahani 2011). This work deserved addressing in both the introduction and discussion.

AUTHORS’ REPLY, point 3:

We added further citations on the interaction between stimulus features (e.g., intensity) and perceived duration, for several sensory modalities, including the visual system (Berglund et al., 1969; Ekman et al., 1969; Kanai et al., 2006; Stévens and Hall, 1966) to the Introduction (line 65) and to the Discussion (line 502). Next to the scalar expectancy theory (Gibbon) that is already mentioned in our manuscript we now also added a citation to the model suggested by Ahrens and Sahani (2011).

Reviewer #3, point 4:

4. Unfortunately, the results from the optogenetic manipulations, in particular those using eNphR3.0 to reduce firing rate, are not entirely convincing. The model presented by the authors makes a specific prediction in terms of the effect of optogenetic manipulation on the psychometric function - changes in the total number of spikes should change in the point of subjective equality (PSE). Based on the error bars presented in the figures, it appears that the effects are not significant (also, please see minor issues below). Traditionally, one would improve the statistical power by using the fit of a logistic function to the psychometric data across data points to measure the shift in PSE. This appears to have not been sufficient, so the authors chose to perform an SVD analysis. It is not clear to the reviewer that this still correctly identifies only shifts in the psychometric function, and not more complicated effects that include shifting, changes in slope, differences in lapse rates, etc. I have two specific requests: first, simulate data where optogenetic manipulations generate different combinations of changes in PSE, slope, and lapse rate and then demonstrate that their method of assessing the significance of changes in behavior induced by optogenetic manipulations are only sensitive to the changes in PSE. Second, please include a summary of parameter fits

of the model to allow the reader to assess how optogenetic manipulation of spike counts influenced the psychometric function.

AUTHORS' REPLY, point 4:

With regard to the results not being “convincing,” we present again some of our responses to Reviewer #1. We note that the effect size of Chr2 in the present study is comparable to the effects reported elsewhere. For instance, one landmark, highly-cited paper (Hanks et al., 2006, *Nature Neuroscience*, 9: 682–689), used electrical microstimulation (a more invasive technique than optogenetics) in monkey LIP during a memory-guided saccade task to support the claim that “microstimulation of this cluster caused an increase in the proportion of choices toward the RF of the stimulated neurons. These results demonstrate that the discharge of LIP neurons is causally related to decision formation in the discrimination task.” Their psychometric curve effect, which has since been taken as unquestioned proof of LIP’s role in visuomotor decision making, is shown below in (left and center plots) comparison to the curve shift from the present manuscript (right plot).

Effect size can be quantified as the mean difference in choice for the stimulation vs the control condition. Using the Hanks et al. data, we compute an effect size of 2.9% for one monkey and 5.4% for the other. In our data, the same measure gives an effect size of 5.1% averaged across 5 rats.

If we use only the central point, corresponding to the condition in which an unbiased subject will split their choices evenly ($T_2 = T_1$ for our study and random motion direction for Hanks et al.), we compute an effect size of 2.2% for one monkey and 5.0% for the other. In our data, the same measure gives an effect size of 11.9% averaged across 5 subjects.

Similar exercises for other published work, including recent optogenetic studies in high-profile journals, confirm that the effect sizes in the present manuscript are not unduly small, especially considering that the aims were not to disrupt behavior but to subtly tune choices. In the revised manuscript, we have emphasized that the optogenetic perturbations left the trained behavior intact (lines 217-221):

If optogenetic manipulation had disrupted performance directly at the decisional level, it would be expected to affect choices no matter which stimulus pair was presented; instead, the manipulations merely shifted the percept in a systematic way, as borne out by the psychometric parameters (Supplementary Fig. S4).

We agree that the effect of optogenetic inhibition on rats’ duration judgment is comparatively small and have revised the manuscript’s organization and interpretations accordingly. In more detail, because

- (a) Halorhodopsin (eNpHR3.0) is a slow active-pump opsin, and is known to require stronger light power as compared to that required to achieve Chr2 excitation effects (Chr2: Boyden et al., 2005; eNpHR3.0: Gradinaru et al., 2010), and
- (b) the maximum light power of the orange (eNpHR3.0) LED was lower than that of the blue (Chr2) LED,

we now consider it more appropriate to present the Halorhodopsin condition as a control rather than (as in the original submission) an active intervention for direct comparison to ChR2. Halorhodopsin is a less potent intervention, unlikely to generate effects of equivalent size.

However, the orange LED optic input could have produced tissue heating and increased neuronal firing; in that case, the perceived stimulus duration might have increased. Instead, the Halorhodopsin condition led to a systematic trend for a *reduced* stimulus duration percept, albeit statistically weak. The value of the Halorhodopsin experiment must be seen, therefore, as a contrast to and control for Chr2 rather than as a principal positive finding. As such, we have moved some parts of those results to Supplementary sections and have presented the ChR2 condition as the main result supporting the hypothesis of vS1 involvement in stimulus duration perception. Details on this reorganization are provided later in the context of other comments. We are grateful for the Reviewer's attention to the Halorhodopsin results, which led us to put them in the proper framework.

As suggested, we further quantified the psychometric curve fitting parameters (new Supplementary Figure S4), in which we could find a significant difference in the bias parameter (μ) between the condition vS1 excitation during stimulus 1 versus vS1 excitation during stimulus 2. For the other fitting parameters, slope (σ) and the two lapse parameters (γ , λ) no significance was detected. When comparing the fitting parameters for vS1 inhibition during stimulus 1 versus vS1 inhibition during stimulus 2 we could not identify significance in any of the parameters, whereas mu parameter shows a trend opposite to vS1 excitation. As discussed before, due to experimental limitations, we could only identify weak perceptual effects for vS1 inhibition, therefore we now show the inhibition results more in the sense of a control experiment. Whereas, when merely comparing rat's choices, at identical stimulus 1 and 2 duration are identical ($\Delta T = 0$, both 334 ms), we can isolate a significant choice bias, also with optogenetic vS1 inhibition (old Figure 2c, middle panel; New Supplementary Figure S2a).

Parameter fits of the model are given in Table 1 (now Supplementary Table T1); that is now mentioned in the Methods (line 966-967).

Reviewer #3, point 5

Minor Issues

The methods for the stimuli and their generation are not entirely clear. At what rate were velocities sampled (e.g. once a ms)? Reporting the range of intensity standard deviations and durations is not sufficient when the values are not spaced regularly. Please be exact about the values used or the method used to select them.

AUTHORS' REPLY, point 5:

In the Methods sentences on stimulus generation, we now added a citation referring to an earlier paper with the identical settings, where they are explained in more detail (Fassihi et al. 2014). We now further added (line 577) the sampling of the velocities (1 kHz). The values of intensities and durations applied are spaced logarithmically and are presented in Supplementary figure S1 (line 594-595).

Reviewer #3, point 6

Paragraph starting at line 117 - it would be more straightforward to just to state that bias was measured as the average change in perceived duration (intensity) as a function of the change in intensity (duration). Using 'likelihood' here suggests, at least to the reviewer, that a Bayesian analysis was performed.

AUTHORS' REPLY, point 6:

We agree with the Reviewer's comment and changed "likelihood" to "average" at the suggested positions.

Reviewer #3, point 7

A supplementary figure mimicking Figure 2A, left and Figure 2A, upper right, but to show the expression of eNpHR3.0 would be appropriate so the reader can assess the quality of transfection and the potential for off-site effects.

AUTHORS' REPLY, point 7:

As mentioned above, we now have treated the results coming from the two rats, transfected with eNpHR3.0, as a control condition. For that reason, we believe this question became less relevant.

Reviewer #3, point 8

Figure 2C and 2D, middle: how, specifically, was the permutation test performed? This should be clearly explained in the methods.

AUTHORS' REPLY, point 8:

We added a more detailed explanation into the Methods (lines 749-766):

To quantify the effect of optogenetic manipulation of vS1 on duration and intensity perception, we resampled the original data set to create 1,000 sets of statistically comparable data, with the same trial size as the original data, allowing for replacement. Each set of resampled responses was parametrized by fitting the logistic function of Equation (3). A support vector machine (SVM) classifier (MATLAB, fitcsvm function) quantified the linear separation between data points (PSE, acquired from the fitted curve versus averaged percentage of choice of the resampled data, not including fitting) with and without optogenetic intervention, making use of 10-fold cross validation to measure the classification error. Specifically, the data were partitioned into 10 random sets. Then, 9 of these were used to train an SVM classifier and the remaining set served as a test. This procedure was repeated 10 times and the statistics for each repetition were combined, giving the rightmost plots of Fig. 2b-d. Statistical significance of the bias in duration and intensity judgements between the different condition pairs (Fig. 2b-e, middle) was tested by subtracting at each resampling iteration the averaged percentage of choice (excluding "easy" trials) between the respective condition pair [$\text{bias}_{\text{it}} = \text{mean}(\% \text{ choicestim2} > \text{stim1} \text{ condition 1}) - \text{mean}(\% \text{ choicestim2} > \text{stim1} \text{ condition 2})$]. p-value was estimated by $(\# \text{ of iterations with bias } \geq 0) / 1000$. To uncover optogenetic effects when not overridden by strong sensory evidence, statistical significance in Supplementary Fig. S2a (middle panel) was tested on the stimulus pairs where $\Delta T = 0$ (for duration rats).

Reviewer #3, point 9

Figure 2C and 2D, middle: was the standard error measured across rats?

AUTHORS' REPLY, point 9:

Yes, the SEM was measured across rats. We now added this information to the Figure 2 captions.

Reviewer #3, point 10

The method for performing the SVD should be more specifically laid out in the methods. The reviewer assumes that only the estimates of the PSE and choice percentage data estimated from resampled data went into the classifier, but that should be clearly stated in the methods.

AUTHORS' REPLY, point 10:

We agree and added this information to the Methods, highlighted in blue (lines 751-756):

... Each set of resampled responses was parametrized by fitting the logistic function of Equation (3). A support vector machine (SVM) classifier (MATLAB, fitcsvm function) quantified the linear separation between data points (PSE, acquired from the fitted curve versus averaged percentage of choice of the resampled data, not including fitting) with and without optogenetic intervention, making use of 10-fold cross validation to measure the classification error.

Reviewer #3, point 11

Figure 2E, middle: are the error bars plotted in this figure?

AUTHORS' REPLY, point 11:

No, we did not include error bars here for clarity, however error bars are shown in the respective curves of the conditions, for instance vS1 excitation during stimulus 2. Now, as old panel 2e moved up to 2b, we also added the resampling points of PSE and percentage of choice to include statistics in the right panel.

Reviewer #3, point 12

Figure 3B: as plot, this figure implies that the PSTH for each neuron was calculated for identical rod velocity traces (e.g. 'frozen noise' trials). If this was not the case, I suggest changing how the traces are presented, or at least clarifying in the legend

AUTHORS' REPLY, point 12:

We thank the reviewer for pointing out this imprecision. The Methods section now clearly explains that multiple stimulus "seeds" drawn from the same statistical distribution were generated as the stimulus set for each intensity and duration. We have also clarified now in the Figure 3b caption that the plate speed traces above represent only one example noisy vibration, whereas the neuronal data consists of an average across varied noisy vibrations. See (lines 360-362):

... **b.** Normalized response of all neurons ($n = 250$) on trials with stimulus 1 and 2 of 334- and 694-ms duration, respectively; intensity 64 mm/s. Above the response plots, example vibrations (plate speed across time) are illustrated as black lines.

Reviewer #3, point 13

Have the authors considered the effect of correlated variability in the spike counts of neurons? That is, when one neuron tends to fire more spikes, other neurons also tend to fire more spikes (e.g. Zohary et. al. 1994). Do they think that this could explain the fact that variability of behavioral biases was much larger than the neurometric biases of their model?

AUTHORS' REPLY, point 13:

Correlated variability in the spike counts of neurons is in fact one of a multitude of factors possibly leading to behavioral variability (e.g. trial history effects, distraction, motivation). We are now collecting larger data sets to try to carry out analyses to account for trial-by-trial choices.

Reviewer #3, point 14

Line 455: I don't think it is necessary to cite a COSYNE abstract when there are plenty of published studies on the neural mechanisms of "empty" interval timing (e.g. Mauk, Buonomano, Merchant, Jazayeri groups).

AUTHORS' REPLY, point 14:

We removed the COSYNE citation and added published studies on the neural mechanism of “empty” interval timing (Jazayeri and Shadlen, 2015; Mauk and Buonomano, 2004; Merchant et al., 2013).

Reviewer #3, point 15

Line 484: It would be appropriate to cite the work on the effect of the order of interval presentations on time perception here (e.g. Dyjas, Bausenhart, and Ulrich).

AUTHORS' REPLY, point 15:

We added the citation to the paper of Bausenhart et al. (2015) to our statements, now line 494.

Sincerely,
Seth Egger

Reference list

- Ahrens, M.B., Sahani, M., 2011. Observers Exploit Stochastic Models of Sensory Change to Help Judge the Passage of Time. *Current Biology* 21, 200–206. <https://doi.org/10.1016/j.cub.2010.12.043>
- Akrami, A., Kopec, C.D., Diamond, M.E., Brody, C.D., 2018. Posterior parietal cortex represents sensory history and mediates its effects on behaviour. *Nature* 554. <https://doi.org/10.1038/nature25510>
- Allman, M.J., DeLeon, I.G., Wearden, J.H., 2011. Psychophysical Assessment of Timing in Individuals With Autism. *Am J Intellect Dev Disabil* 116. <https://doi.org/10.1352/1944-7558-116.2.165>
- Andrei, A.R., Debes, S., Chelaru, M., Liu, X., Rodarte, E., Spudich, J.L., Janz, R., Dragoi, V., 2021. Heterogeneous side effects of cortical inactivation in behaving animals. *Elife* 10. <https://doi.org/10.7554/eLife.66400>
- Balasubramaniam, R., Haegens, S., Jazayeri, M., Merchant, H., Sternad, D., Song, J.-H., 2021. Neural Encoding and Representation of Time for Sensorimotor Control and Learning. *The Journal of Neuroscience* 41. <https://doi.org/10.1523/JNEUROSCI.1652-20.2020>
- Bausenhart, K.M., Dyjas, O., Ulrich, R., 2015. Effects of stimulus order on discrimination sensitivity for short and long durations. *Atten Percept Psychophys* 77, 1033–1043. <https://doi.org/10.3758/s13414-015-0875-8>
- Berglund, B., Berglund, U., Ekman, G., Frankehaeuser, M., 1969. THE INFLUENCE OF AUDITORY STIMULUS INTENSITY ON APPARENT DURATION. *Scand J Psychol* 10, 21–26. <https://doi.org/10.1111/j.1467-9450.1969.tb00003.x>

- Boyden, E.S., Zhang, F., Bamberg, E., Nagel, G., Deisseroth, K., 2005. Millisecond-timescale, genetically targeted optical control of neural activity. *Nat Neurosci* 8, 1263–1268.
<https://doi.org/10.1038/nn1525>
- BROCK, J., BROWN, C.C., BOUCHER, J., RIPPON, G., 2002. The temporal binding deficit hypothesis of autism. *Dev Psychopathol* 14. <https://doi.org/10.1017/S0954579402002018>
- Brown, S.W., 1997. Attentional resources in timing: Interference effects in concurrent temporal and nontemporal working memory tasks. *Percept Psychophys* 59, 1118–1140.
<https://doi.org/10.3758/BF03205526>
- Bruno, R.M., 2011. Synchrony in sensation. *Curr Opin Neurobiol* 21, 701–708.
<https://doi.org/10.1016/j.conb.2011.06.003>
- Cardin, J.A., Carlén, M., Meletis, K., Knoblich, U., Zhang, F., Deisseroth, K., Tsai, L.-H., Moore, C.I., 2010. Targeted optogenetic stimulation and recording of neurons in vivo using cell-type-specific expression of Channelrhodopsin-2. *Nat Protoc* 5, 247–254. <https://doi.org/10.1038/nprot.2009.228>
- Ekman, G., Frankenhaeuser, M., Berglund, B., Waszak, M., 1969. Apparent Duration as a Function of Intensity of Vibrotactile Stimulation. *Percept Mot Skills* 28, 151–156.
<https://doi.org/10.2466/pms.1969.28.1.151>
- Esmaili, V., Diamond, M.E., 2019. Neuronal Correlates of Tactile Working Memory in Prefrontal and Vibrissal Somatosensory Cortex. *Cell Rep* 27, 3167-3181.e5.
<https://doi.org/10.1016/j.celrep.2019.05.034>
- Fassihi, A., Akrami, A., Esmaili, V., Diamond, M.E., 2014. Tactile perception and working memory in rats and humans. *Proc Natl Acad Sci U S A* 111. <https://doi.org/10.1073/pnas.1315171111>
- Gradinaru, V., Zhang, F., Ramakrishnan, C., Mattis, J., Prakash, R., Diester, I., Goshen, I., Thompson, K.R., Deisseroth, K., 2010. Molecular and Cellular Approaches for Diversifying and Extending Optogenetics. *Cell* 141. <https://doi.org/10.1016/j.cell.2010.02.037>
- Hanks, T.D., Ditterich, J., Shadlen, M.N., 2006. Microstimulation of macaque area LIP affects decision-making in a motion discrimination task. *Nat Neurosci* 9, 682–689. <https://doi.org/10.1038/nn1683>
- Honma, M., Itoi, C., Midorikawa, A., Terao, Y., Masaoka, Y., Kuroda, T., Futamura, A., Shiromaru, A., Ohta, H., Kato, N., Kawamura, M., Ono, K., 2019. Contraction of distance and duration production in autism spectrum disorder. *Sci Rep* 9. <https://doi.org/10.1038/s41598-019-45250-8>
- Jazayeri, M., Shadlen, M.N., 2015. A Neural Mechanism for Sensing and Reproducing a Time Interval. *Current Biology* 25. <https://doi.org/10.1016/j.cub.2015.08.038>
- Kanai, R., Paffen, C.L.E., Hogendoorn, H., Verstraten, F.A.J., 2006. Time dilation in dynamic visual display. *J Vis* 6, 8. <https://doi.org/10.1167/6.12.8>
- Li, H., Wang, J., Liu, G., Xu, J., Huang, W., Song, C., Wang, D., Tao, H.W., Zhang, L.I., Liang, F., 2021. Phasic Off responses of auditory cortex underlie perception of sound duration. *Cell Rep* 35.
<https://doi.org/10.1016/j.celrep.2021.109003>

- Matell, M.S., Meck, W.H., Nicolelis, M.A.L., 2003. Interval timing and the encoding of signal duration by ensembles of cortical and striatal neurons. *Behavioral Neuroscience* 117, 760–773. <https://doi.org/10.1037/0735-7044.117.4.760>
- Mauk, M.D., Buonomano, D. V., 2004. THE NEURAL BASIS OF TEMPORAL PROCESSING. *Annu Rev Neurosci* 27, 307–340. <https://doi.org/10.1146/annurev.neuro.27.070203.144247>
- MCHAFFIE, J., STANFORD, T., STEIN, B., COIZET, V., REDGRAVE, P., 2005. Subcortical loops through the basal ganglia. *Trends Neurosci* 28, 401–407. <https://doi.org/10.1016/j.tins.2005.06.006>
- Meck, W.H., Benson, A.M., 2002. Dissecting the brain's internal clock: How frontal-striatal circuitry keeps time and shifts attention. *Brain Cogn* 48. <https://doi.org/10.1006/brcg.2001.1313>
- Merchant, H., Harrington, D.L., Meck, W.H., 2013. Neural Basis of the Perception and Estimation of Time. *Annu Rev Neurosci* 36, 313–336. <https://doi.org/10.1146/annurev-neuro-062012-170349>
- Mikulovic, S., Pupe, S., Peixoto, H.M., Do Nascimento, G.C., Kullander, K., Tort, A.B.L., Leão, R.N., 2016. On the photovoltaic effect in local field potential recordings. *Neurophotonics* 3, 015002. <https://doi.org/10.1117/1.NPh.3.1.015002>
- Monteiro, T., Rodrigues, F.S., Pexirra, M., Cruz, B.F., Gonçalves, A.I., Rueda-Orozco, P.E., Paton, J.J., 2023. Using temperature to analyze the neural basis of a time-based decision. *Nat Neurosci* 26, 1407–1416. <https://doi.org/10.1038/s41593-023-01378-5>
- Murray, J.M., Escola, G.S., 2017. Learning multiple variable-speed sequences in striatum via cortical tutoring. *Elife* 6. <https://doi.org/10.7554/eLife.26084>
- Owen, S.F., Liu, M.H., Kreitzer, A.C., 2019. Thermal constraints on in vivo optogenetic manipulations. *Nat Neurosci* 22. <https://doi.org/10.1038/s41593-019-0422-3>
- Penney, T.B., Gibbon, J., Meck, W.H., 2000. Differential effects of auditory and visual signals on clock speed and temporal memory. *J Exp Psychol Hum Percept Perform* 26, 1770–1787. <https://doi.org/10.1037/0096-1523.26.6.1770>
- Raviv, O., Ahissar, M., Loewenstein, Y., 2012. How Recent History Affects Perception: The Normative Approach and Its Heuristic Approximation. *PLoS Comput Biol* 8. <https://doi.org/10.1371/journal.pcbi.1002731>
- Stévens, J.C., Hall, J.W., 1966. Brightness and loudness as functions of stimulus duration. *Percept Psychophys* 1, 319–327. <https://doi.org/10.3758/BF03215796>
- Stevenson, R.A., Park, S., Cochran, C., McIntosh, L.G., Noel, J.P., Barense, M.D., Ferber, S., Wallace, M.T., 2017. The associations between multisensory temporal processing and symptoms of schizophrenia. *Schizophr Res* 179. <https://doi.org/10.1016/j.schres.2016.09.035>
- Toso, A., Fassihi, A., Paz, L., Pulecchi, F., Diamond, M.E., 2021a. A sensory integration account for time perception. *PLoS Comput Biol* 17. <https://doi.org/10.1371/JOURNAL.PCBI.1008668>
- Toso, A., Reinartz, S., Pulecchi, F., Diamond, M.E., 2021b. Time coding in rat dorsolateral striatum. *Neuron*. <https://doi.org/10.1016/j.neuron.2021.08.020>

- Ueda, N., Maruo, K., Sumiyoshi, T., 2018. Positive symptoms and time perception in schizophrenia: A meta-analysis. *Schizophr Res Cogn* 13. <https://doi.org/10.1016/j.scog.2018.07.002>
- Wolff, S.B., Ölveczky, B.P., 2018. The promise and perils of causal circuit manipulations. *Curr Opin Neurobiol* 49. <https://doi.org/10.1016/j.conb.2018.01.004>
- Zakay, D., Block, R.A., 1996. The role of attention in time estimation processes. pp. 143–164. [https://doi.org/10.1016/S0166-4115\(96\)80057-4](https://doi.org/10.1016/S0166-4115(96)80057-4)
- Znamenskiy, P., Zador, A.M., 2013. Corticostriatal neurons in auditory cortex drive decisions during auditory discrimination. *Nature* 497, 482–485. <https://doi.org/10.1038/nature12077>

REVIEWER COMMENTS

Reviewer #1 (Remarks to the Author):

The manuscript is much improved in terms of clarity and experimental/analytical support for the main conclusions. However, it is not useful to cite previous papers with small effect size as 'gold' standard for acceptable effect sizes. And it is certainly not true that the small effect size in the cited paper is taken as unquestioned proof of LIP's role in visuomotor DM. As a field, we should collectively hold ourselves for higher, replicatable standards, rather than using others not meeting the standards as justifications. Nevertheless, we appreciate the further statistical analyses (psychometric parameters etc) conducted for the optogenetic experiments.

Most of our concerns have been addressed. Below are the two remaining issues/suggestions.

1. The authors proposed 2 different codes for intensity (rate code) and duration (count integration code) coding in S1. What happens when these two codes conflict with each other, especially for the duration rats? For example, sometimes rats have to compare high intensity short duration stimuli with low intensity long duration stimuli. If high intensity leads to high rates in S1, it will also result in higher overall count, which could conflict with the short duration count code? We refer to these stimuli as incongruent.

In Fig. 3e,f, the authors compared rate versus count coding of duration for the duration rats, averaged across different intensity trial types. Can the authors separate trial types into congruent and incongruent stimuli? It seems like the count code can be compromised in the incongruent stimuli if the rate coding of intensity also exists in these rats, even though intensity is not behaviourally relevant. Similarly, it would be interesting to look at count coding of intensity in intensity rats as well?

Intuitively, count coding of duration should not interfere with intensity rate coding but intensity rate coding would interfere with duration count coding, especially in incongruent trials. I also wonder if that may be the reasons why duration rats' performance in duration rule is worse than intensity rats' performance for the intensity rule. How would the authors incorporate these in the modelling?

2. There has been some controversy in terms of whether variability in the dorsal medial striatum contributes to time perception. The authors have previously found that DMS is not required for judgment of stimulus duration in this task, and are now suggesting that S1 is directly involved. Can the authors further discuss what aspects of the tasks may be able to explain such discrepancies across groups (regarding contributions of different brain areas to interval judgement)?

Reviewer #2 (Remarks to the Author):

Dear Authors,

thanks for the detailed reply to my previous comments. I am not completely convinced by all the arguments that the authors put forward. For instance the fact that science proceeds in small steps to me is a manifesto of "incremental science" hence argues in favour of my take which is that the work lacks breakthrough potential. Also the fact that the current manipulations affect duration rather than onset-

offset is also to be expected given that most literature on the effect of stimulus intensity reports multiplicative effects, proportional to the interval.

However amongst the various things that the authors argue one is in itself novel enough. Given that there are multiple sensory pathways from vibrissae to cortex, the fact that manipulation of vS1 alone is sufficient to mimick the effects of time perception is notable. Hence I suggest to change the intro to emphasize better this aspect which currently is quite subdued.

So all in all the breakthrough potential issue can get a pass mark.

As for the rest I would just like to stress the right hand side scatter plots of Figures 2b-e (or for instance Figure S5), contain several issues. Firstly it is not clear what the Y axis is. Is it overall proportion of judgments $T2 > T1$ or only at $\Delta = 0$? If it is the former, I would like to stress that the measure is highly dependent on the range of stimuli which are delivered so it is not quite as distilled as one would wish. That aside there is the risk of double dipping: both the Y-axis and the X-axis are statistics trying to capture the same first order trend in the data. Y axis is the height of the datapoints of the central psychometric curves, X axis is the left-right shift of the sigmoidal, monotonic fit to the data. Indeed it is nearly impossible to have a condition that yields higher proportions $T2 > T1$ and still the fit tends towards a positive shift. For this reason I believe that the presentation (and even worse the analysis) of these data is misleading. Also it bears no added value as the psychometric function fits alone provide sufficient unbiased information. Hence they should be removed

Reviewer #3 (Remarks to the Author):

The authors have sufficiently addressed my concerns and I believe the manuscript will be acceptable for publication following careful editing for grammatical errors.

Dear Editor and Reviewers,

Below, please find the point-by-point replies to the second round of review. As for the first round, the reviewer comments are in blue font and our replies are in black font. In the resubmitted manuscript, revised or new text with respect to the revised submission are in dark blue font.

REVIEWER COMMENTS

Reviewer #1 (Remarks to the Author),

The manuscript is much improved in terms of clarity and experimental/analytical support for the main conclusions. However, it is not useful to cite previous papers with small effect size as 'gold' standard for acceptable effect sizes. And it is certainly not true that the small effect size in the cited paper is taken as unquestioned proof of LIP's role in visuomotor DM. As a field, we should collectively hold ourselves for higher, replicatable standards, rather than using others not meeting the standards as justifications. Nevertheless, we appreciate the further statistical analyses (psychometric parameters etc) conducted for the optogenetic experiments.

Most of our concerns have been addressed. Below are the two remaining issues/suggestions.

AUTHORS' REPLY

We agree that a comparison of the effect sizes in our study to those of previous studies is not useful within the body of the paper. Such comparisons were restricted to our reply letter; the manuscript itself contains no such comparison. We fully agree that the effects described in this study need to be robust and must stand on their own two feet, without comparison to earlier studies. We are confident that the claims in the current version are supported by solid evidence.

Reviewer #1, point 1:

1. The authors proposed 2 different codes for intensity (rate code) and duration (count integration code) coding in S1. What happens when these two codes conflict with each other, especially for the duration rats? For example, sometimes rats have to compare high intensity short duration stimuli with low intensity long duration stimuli. If high intensity leads to high rates in S1, it will also result in higher overall count, which could conflict with the short duration count code? We refer to these stimuli as incongruent.

AUTHORS' REPLY, point 1:

This point perfectly targets the main finding of the study, the multiplexing of both codes within the same vS1 population. Let us begin with the question, "What happens when these two codes conflict with each other, especially for the duration rats"? The answer is that the rat is biased and can make errors as a consequence. This is apparent, for instance in Figure 1c, below (in the

resubmitted revision, we have relabeled the axes to make the plots easier to interpret). It is seen that Duration rats were able to successfully compare $T2$ to $T1$ (upper plot, in gray), however the intensity of the vibration caused these same rats to be biased in their choices (lower plot, in gray). If intensity had *not* created a bias, the slope of the gray line in the lower plot would be 0. (Just as the bias evoked by stimulus duration in judging intensity is given by the slope of the red line in the upper plot.) This bias occurred, in our framework, because the coding of one percept was not fully separable from the coding of the other percept. For example, whenever a Duration rat must judge a short, strong stimulus, the high firing rate caused by its intensity causes the integrated spike quantity to be large – the short stimulus thus will be judged as having a longer duration than if the same duration were accompanied by low intensity. In the lower plot below, when stimulus 2 is of high intensity, making ΔI positive and large, stimulus 2 is more likely to be judged of longer duration than stimulus 1, giving a greater % choice $T2 > T1$. This is the bias embodied in the positive slope of the gray line of the lower plot.

c

To the further question of the Reviewer (“For example, sometimes rats have to compare high intensity short duration stimuli with low intensity long duration stimuli. If high intensity leads to high rates in S1, it will also result in higher overall count, which could conflict with the short duration count code?”), the answer is **Yes**. A more detailed reply continues below.

Reviewer #1, point 1 (CONTINUED):

In Fig. 3e,f, the authors compared rate versus count coding of duration for the duration rats, averaged across different intensity trial types. Can the authors separate trial types into congruent and incongruent stimuli? It seems like the count code can be compromised in the incongruent stimuli if the rate coding of intensity also exists in these rats, even though intensity is not behaviourally relevant. Similarly, it would be interesting to look at count coding of intensity in intensity rats as well?

AUTHORS' REPLY, point 1 (CONTINUED):

We have added new Supplementary Figure S14 to unpack the interesting problem delineated by the Reviewer. Congruent and incongruent trials are best sorted out from behavioral rather than physiological data because there are many more trials and more stimulus combinations. Because it is established that higher and lower intensity are represented by higher and lower vS1 firing rate, respectively (Figure 3d), and because vS1 firing lasts longer as stimulus duration increases (Figure 3h), the problem can be addressed even without a direct neuronal firing analysis. This new analysis substantiates the Reviewer's expectation that the degree of congruence of the intensity feature affects the perceived duration, and the degree of congruence of the duration feature affects the perceived intensity, consistent with the multiplexing hypothesis. The analysis is described in the following new text and figure (lines 415-441):

The model predicts that when the vS1 firing evoked by the vibration's duration and intensity is integrated, the two stimulus features will be either congruent or incongruent in their contribution to perceived duration. To test this, using the stimulus generalization matrix given in Supplementary Fig. S1, we identified two groups of trials. For Duration rats, congruent trials (group 1) were characterized by stimulus 2 of short duration (264 ms) and low intensity (34, 42, or 52 mm/s) or else long duration (422 ms) and high intensity (78, 96, or 119 mm/s); incongruent trials (group 2) were characterized by stimulus 2 of short duration and high intensity or else long duration and low intensity (Supplementary Fig. S14). All of these instances of stimulus 2 were judged by the rats in comparison to stimulus 1, which had intermediate duration (334 ms) and intermediate intensity (64 mm/s). The prediction is that performance for group 1 trials will be better than performance for group 2 trials. This is because the *congruence* of intensity with duration in group 1 (where intensity causes short stimuli to feel shorter or causes long stimuli to feel longer) will lead the perceived duration to be more distant from that of stimulus 1, and will thus make the judgment easier and more accurate. By contrast, the *incongruence* of intensity with duration in group 2 (where intensity causes short stimuli to feel longer or causes long stimuli to feel shorter) will lead the perceived duration to be closer to that of stimulus 1, and will thus make the judgment more difficult and less accurate. Accuracy was 74.1% (STD: +/- 1.26) for group 1 trials, significantly better ($p < 0.001$) than the 64.9 % (STD: +/- 0.76) accuracy for group 2 trials. This analysis substantiates the model's prediction that the intensity feature can act congruently or incongruently with the duration feature.

Although the present study focuses on duration perception, we verified the analogous effect in Intensity rats (Supplementary Fig. S14). This shows that, in general, the congruence/incongruence of the irrelevant feature acts on both percepts, supporting the framework of multiplexed coding of distinct percepts. An additional treatment of the duration/intensity confound in Intensity rats can be found in (Fassihi et al., 2017; Toso et al., 2021).

Reviewer #1, point 1 (CONTINUED):

Intuitively, count coding of duration should not interfere with intensity rate coding but intensity rate coding would interfere with duration count coding, especially in incongruent trials. I also wonder if that may be the reasons why duration rats' performance in duration rule is worse than intensity rats' performance for the intensity rule. How would the authors incorporate these in the modelling?

AUTHORS' REPLY, point 1 (CONTINUED):

Our model is only partly in agreement with the predictions above. We break them into two separate predictions.

1. "intensity rate coding would interfere with duration count coding". This agrees with our model. Although the linearly summated spike count is tested for plausibility as a correlate of perceived duration in Figure 3f, g, i, the full model argues for nonlinearly integrated and accumulated spike count. Going from qualitative tests to computational modeling demonstrates (Figure 4) that the duration percept could be recovered by positing a leaky integration of vS1 firing with time constant τ of 990 ms (Supplementary Table T1). This mechanism can explain the empirical results of Figure 1, discussed above, where a longer vibration is judged as longer (as it must) yet a *stronger* vibration is also judged as longer. This is because the higher vS1 firing rate evoked by the strong stimulus leads to a greater integrated quantity.

2. "count coding of duration should not interfere with intensity rate coding". This does not agree with our model (also see Supplementary Figure S14). Though not the focus of the current study, previous work (Fassihi et al., 2017; Toso et al., 2021) has considered the neuronal coding that underlies the perception of intensity, through neuronal population analyses, psychophysics, and computational modeling. This earlier work indicated that the final perceived intensity of a vibration involves not the *instantaneous* firing rate, but an *integrated, accumulated* quantity. The intensity perception model is similar in structure to that of Figure 4 of the present work. However, fitting the modeling parameters to psychophysical and neuronal data showed that intensity percepts could be recovered by positing a leaky integration of vS1 firing with time constant τ of 70-100 ms, much shorter than the τ for duration perception. Additionally, the intensity accumulator gives strong weight to input from intensity-coding vS1 neurons, while the duration accumulator gives less weight to input from intensity-coding vS1 neurons. For this reason, and perhaps counterintuitively, the experimental results of Figure 1 show that a longer stimulus is perceived as stronger. This is the finding that disagrees with the Reviewer's expectation that count coding of duration should not interfere with intensity rate coding. In short, vS1 firing is integrated as part of both percepts.

As to why duration rats' performance in the duration rule is worse than intensity rats' performance for the intensity rule, it is hard to provide a definitive answer based on available data. The neuronal populations that encode the two explicit percepts, beyond the multiplexing stage of vS1, might have different functional properties. Even for human subjects, duration delayed comparison is more challenging than intensity delayed comparison.

Reviewer #1, point 2:

2. There has been some controversy in terms of whether variability in the dorsal medial striatum contributes to time perception. The authors have previously found that DMS is not required for judgment of stimulus duration in this task, and are now suggesting that S1 is directly involved. Can the authors further discuss what aspects of the tasks may be able to explain such discrepancies across groups (regarding contributions of different brain areas to interval judgement)?

AUTHORS' REPLY, point 2:

The results of the present manuscript do not rule out DMS as part of the time perception and decision making circuitry. Our earlier results suggested that DMS activity represents the upcoming choice, more so than the explicit real-time sensation of elapsed time. One of the key arguments in that work is that duration perception is biased by stimulus intensity (see point 1, above), while the DMS representation of time only appeared to be biased by stimulus intensity in dimensions (time points, correlations) that could not be separated from an expected representation of the upcoming choice.

The results in the present manuscript, showing direct involvement of vS1 in judgements of stimulus duration, would still fit with DMS being an area involved in the time dependent decision making circuitry. Because the current results do not specify the potential role of DMS beyond what is already documented in the literature, we prefer not to expand that discussion.

Reviewer #2 (Remarks to the Author),

Dear Authors,

thanks for the detailed reply to my previous comments. I am not completely convinced by all the arguments that the authors put forward. For instance the fact that science proceeds in small steps to me is a manifesto of "incremental science" hence argues in favour of my take which is that the work lacks breakthrough potential. Also the fact that the current manipulations affect duration rather than onset-offset is also to be expected given that most literature on the effect of stimulus intensity reports multiplicative effects, proportional to the interval.

However amongst the various things that the authors argue one is in itself novel enough. Given that there are multiple sensory pathways from vibrissae to cortex, the fact that manipulation of vS1 alone is sufficient to mimick the effects of time perception is notable. Hence I suggest to change the intro to emphasize better this aspect which currently is quite subdued.

So all in all the breakthrough potential issue can get a pass mark.

AUTHORS' REPLY:

Thank you for this summary. In the Introduction we have now emphasized the importance of establishing which structures within sensory pathways are components of the neuronal representation of duration perception (lines 65-66):

As such, it is useful to pinpoint which sensory processing structure participates in the percept.

Reviewer #2, point 1:

As for the rest I would just like to stress the right hand side scatter plots of Figures 2b-e (or for instance Figure S5), contain several issues. Firstly it is not clear what the Y axis is. Is it overall proportion of judgments $T2 > T1$ or only at $\Delta=0$? If it is the former, I would like to stress that the measure is highly dependent on the range of stimuli which are delivered so it is not quite as distilled as one would wish. That aside there is the risk of double dipping: both the Y-axis and the X-axis are statistics trying to capture the same first order trend in the data. Y axis is the height of the datapoints of the central psychometric curves, X axis is the left-right shift of the sigmoidal, monotonic fit to the data. Indeed it is nearly impossible to have a condition that yields higher proportions $T2 > T1$ and still the fit tends towards a positive shift. For this reason I believe that the presentation (and even worse the analysis) of these data is misleading. Also it bears no added value and the psychometric function fits alone provide sufficient unbiased information. Hence they should be removed.

AUTHORS' REPLY, point 1:

The measures of $T2 > T1$ in the scatter plots do not merely represent the decisions in the trials with ΔT (or ΔI) = 0, but as in all our bias measures, are an average of the choices to all stimulus durations, apart from the two easiest ones (See Methods lines 771-775).

We agree that the measure of $T2 > T1$ and the PSTH we used for the scatter plots are highly correlated measures, therefore statements about their correlation itself does not add anything. However we added these resampled points to give a more quantitative representation of the data and show the distribution of the resampled points out of two conditions being either totally overlapping or separate. Therefore we believe keeping them still adds relevant information. Since the other Reviewers did not converge to the view of the scatter plots being without value, our preference is to keep them.

Reviewer #3 (Remarks to the Author),

The authors have sufficiently addressed my concerns and I believe the manuscript will be acceptable for publication following careful editing for grammatical errors.

AUTHORS' REPLY:

No further revisions applied.

Reference list

Fassihi, A., Akrami, A., Pulecchi, F., Schönfelder, V., Diamond, M.E., 2017. Transformation of Perception from Sensory to Motor Cortex. *Current Biology* 27. <https://doi.org/10.1016/j.cub.2017.05.011>

Toso, A., Fassihi, A., Paz, L., Pulecchi, F., Diamond, M.E., 2021. A sensory integration account for time perception. *PLoS Comput Biol* 17. <https://doi.org/10.1371/JOURNAL.PCBI.1008668>

REVIEWERS' COMMENTS

Reviewer #1 (Remarks to the Author):

The authors have addressed all my concerns, and I support publication.

Reviewer #2 (Remarks to the Author):

Dear All,

thank you very much for your response. I am not entirely convinced by the author's response but I think overall the manuscript as it is now presents a balanced view of the results so I have no further concerns.